# Effects of ocean acidification on pelagic carbon fluxes in a mesocosm experiment

Kristian Spilling[1, 2], Kai G. Schulz[3], Allanah J. Paul[4], Tim Boxhammer[4], Eric P. Achterberg[4, 5], Thomas Hornick[6], Silke Lischka[4], Annegret Stuhr[4], Rafael Bermúdez[4, 7], Jan Czerny[4], Kate Crawfurd[8], Corina P. D. Brussaard[8, 9], Hans-Peter Grossart[6, 10], Ulf Riebesell[4]

[1] {Marine Research Centre, Finnish Environment Institute, P.O. Box 140, 00251 Helsinki, Finland}

[2] {Tvärminne Zoological Station, University of Helsinki, J. A. Palménin tie 260, 10900 Hanko, Finland}

[3] {Centre for Coastal Biogeochemistry, Southern Cross University, Military Road, East Lismore, NSW 2480, Australia}

[4] {GEOMAR Helmholtz Centre for Ocean Research Kiel, Düsternbrooker Weg 20, 24105 Kiel, Germany}

[5] {National Oceanography Centre Southampton, European Way, University of Southampton, Southampton, SO14 3ZH, UK}

[6] {Leibniz Institute of Freshwater Ecology and Inland Fisheries (IGB), Experimental Limnology, 16775 Stechlin, Germany}

[7] {Facultad de Ingeniería Marítima, Ciencias Biológicas, Oceánicas y Recursos Naturales. ESPOL, Escuela Superior Politécnica del Litoral, Guayaquil, Ecuador}

[8] {NIOZ Royal Netherlands Institute for Sea Research, Department of Marine Microbiology and Biogeochemistry, and Utrecht University, P.O. Box 59, 1790 AB Den Burg, Texel, The Netherlands}

[9] {Department of Aquatic Microbiology, Institute for Biodiversity and Ecosystem Dynamics (IBED), University of Amsterdam, The Netherlands

[10] {Potsdam University, Institute for Biochemistry and Biology, 14469 Potsdam, Germany}

Correspondence to: K. Spilling (kristian.spilling@environment.fi)

Running title: Modified pelagic carbon fluxes

Key words: Carbon fluxes, carbon budget, gross primary production, respiration, bacterial production, sinking carbon flux, $CO_2$ exchange with atmosphere

**Abstract**
About a quarter of anthropogenic $CO_2$ emissions are currently taken up by the oceans
decreasing seawater pH. We performed a mesocosm experiment in the Baltic Sea in order to
investigate the consequences of increasing $CO_2$ levels on pelagic carbon fluxes. A gradient of
different $CO_2$ scenarios, ranging from ambient (~370 µatm) to high (~1200 µatm), were set
up in mesocosm bags (~55 m$^3$). We determined standing stocks and temporal changes of total
particulate carbon (TPC), dissolved organic carbon (DOC), dissolved inorganic carbon (DIC)
and particulate organic carbon (POC) of specific plankton groups. We also measured carbon
flux via $CO_2$ exchange with the atmosphere and sedimentation (export); and biological rate
measurements of primary production, bacterial production and total respiration. The
experiment lasted for 44 days and was divided into three different phases (I: *t0-t16*; II: *t17-*
*t30*; III: *t31-t43*). Pools of TPC, DOC and DIC were approximately 420, 7200 and 25200
mmol C m$^{-2}$ at the start of the experiment, and the initial $CO_2$ additions increased the DIC
pool by ~7% in the highest $CO_2$ treatment. Overall, there was a decrease in TPC and increase
of DOC over the course of the experiment. The decrease in TPC was lower, and increase in
DOC higher, in treatments with added $CO_2$. During Phase I the estimated gross primary
production (GPP) was ~100 mmol C m$^{-2}$ d$^{-1}$; from which 75-95% were respired, ~1% ended
up in the TPC (including export) and 5-25% added to the DOC pool. During Phase II, the
respiration loss increased to ~100% of GPP at the ambient $CO_2$ concentration, whereas
respiration was lower (85-95% of GPP) in the highest $CO_2$ treatment. Bacterial production
was ~30% lower, on average, at the highest $CO_2$ concentration compared with the controls
during Phases II and III. This resulted in a higher accumulation DOC standing stock and
lower reduction in TPC in the elevated $CO_2$ treatments at the end of Phase II extending
throughout Phase III. The "extra" organic carbon at high $CO_2$ remained fixed in an increasing
biomass of small-sized plankton and in the DOC pool, and did not transfer into large, sinking
aggregates. Our results revealed a clear effect of increasing $CO_2$ on the carbon budget and
mineralization, in particular under nutrient limited conditions. Lower carbon loss processes
(respiration and bacterial remineralization) at elevated $CO_2$ levels resulted in higher TPC and
DOC pools compared with the ambient $CO_2$ concentration. These results highlight the
importance to address not only net changes in carbon standing stocks, but also carbon fluxes
and budgets to better disentangle the effects of ocean acidification.

## 1    Introduction

Combustion of fossil fuels and change in land use, have caused increasing atmospheric concentrations of carbon dioxide ($CO_2$). Ca. 25% of the anthropogenic $CO_2$ is absorbed by the oceans, thereby decreasing surface water pH, a process termed ocean acidification (Le Quéré et al., 2009). Ocean acidification and its alterations of aquatic ecosystems have received considerable attention during the past decade, but there are many open questions, in particular related to consequences for planktonic mediated carbon fluxes.

Some studies on ocean acidification have reported increased carbon fixation (Egge et al., 2009; Engel et al., 2013), bacterial production (Grossart et al., 2006) and bacterial degradation of polysaccharides (Piontek et al., 2010) at enhanced $CO_2$ levels, with potential consequences for carbon fluxes within pelagic ecosystems and export to the deep ocean, i.e. the biological carbon pump. Increasing carbon fixation in a high $CO_2$ environment can translate into an enhanced sequestration of carbon (Riebesell et al., 2007), but this depends on numerous environmental factors including phytoplankton community composition, aggregate formation and nutrient availability. For example, if the community shifts towards smaller cell sizes and/or enhanced cycling of organic matter carbon, export from the upper water layers may decrease (Czerny et al., 2013a).

The effect of ocean acidification has mostly been studied in marine ecosystems under high phytoplankton biomass. Brackish water has lower buffering capacity than ocean water and the pH fluctuates more. The limited number of studies of ocean acidification in brackish water and indications that ocean acidification effects are greatest under nutrient limitation (De Kluijver et al., 2010), motivated this mesocosm study in the Baltic Sea during low nutrient, summer months.

The Baltic Sea is functionally much like a large estuary, with a salinity gradient ranging from approximately 20 in the South-West to <3 in the Northernmost Bothnian Bay. It is an almost landlocked body of water with a large population in its vicinity (~80 million). Human activities (e.g. agriculture, shipping and fishing) cause a number of environmental problems such as eutrophication and pollution. As a coastal sea projected to change rapidly due to interaction of direct and indirect anthropogenic pressures, the Baltic Sea can be seen as a model ecosystem to study global change scenarios (Niiranen et al., 2013).

Most primary data from this experiment are published in several papers of this Special Issue (Riebesell et al., 2015). The aim of the present paper is to provide an overarching synthesis of

all information related to carbon standing stocks and fluxes. This enabled us to calculate
carbon budgets in relation to different $CO_2$ levels.


**2       Materials and methods**

**2.1.    Experimental set-up**
Six Kiel Off-Shore Mesocosms for future Ocean Simulations (KOSMOS; with a volume of
ca. 55 m$^3$) were moored at Storfjärden, on the south west coast of Finland (59° 51.5' N; 23°
15.5' E) on 12 June 2012 (nine KOSMOS units were originally deployed but three were lost
due to leaks). A more detailed description of the set-up can be found in Paul et al. (2015).
The mesocosms extended from the surface down to 19 m depth and had a conical bottom end,
which enabled quantitative collection of the settling material. Different $CO_2$ levels in the bags
were achieved by adding filtered (50 µm), $CO_2$-saturated seawater. The $CO_2$ enriched water
was evenly distributed over the upper 17 m of the water columns and added in 4 consecutive
time steps ($t0 – t3$). Two controls and four treatments were used, and for the controls, filtered
seawater (without additional $CO_2$ enrichment) was added. The $CO_2$ fugacity gradient after all
additions ranged from ambient (average throughout the experiment: ~370 µatm $fCO_2$) in the
two control mesocosms (M1 and M5), up to ~1200 µatm $fCO_2$ in the highest treatment (M8).
We used the average $fCO_2$ throughout this experiment (from $t1 – t43$) to denote the different
treatments: 365 (M1), 368 (M5), 497 (M7), 821 (M6), 1007 (M3) and 1231 (M8) µatm $fCO_2$.
On $t15$, additional $CO_2$–saturated seawater was added to the upper 7 m in the same manner as
the initial enrichment, to counteract outgassing of $CO_2$.
We sampled the mesocosm every morning, but some variables were determined only every
second day. Depth-integrated water samples (0 – 17 m) were taken by using integrating water
samplers (IWS, HYDRO-BIOS, Kiel). The water was collected into plastic carboys (10 L)
and taken to the laboratory for sub-sampling and subsequent determination of carbon stocks.

**2.2. Primary variables**

For more detailed descriptions of the primary variables and the different methods used during this $CO_2$ mesocosm campaign, we refer to other papers in this joint volume: i.e. total particulate carbon (TPC), dissolved organic carbon (DOC), and dissolved inorganic carbon (DIC) are described by Paul et al. (2015); micro and nanophytoplankton enumeration by Bermúdez et al. (2016); picophytoplankton, heterotrophic prokaryotes and viruses by Crawfurd et al. (2016); zooplankton community by Lischka et al. (2015); primary production and respiration by Spilling et al. (2016); bacterial production (BP) by Hornick et al. (2016); and sedimentation by Boxhammer et al. (2016); and Paul et al. (2015).

Briefly, samples for TPC (500 mL) were GF/F filtered and determined using an elemental analyzer (EuroAE). DOC was measured using the high temperature combustion method (Shimadzu TOC –VCPN) following Badr et al. (2003). DIC was determined by infrared absorption (LI-COR LI-7000 on an AIRICA system). The DIC concentrations were converted from µmol $kg^{-1}$ to µmol $L^{-1}$ using the average seawater density of 1.0038 kg $L^{-1}$ throughout the experiment. Settling particles were quantitatively collected every other day from sediment traps at the bottom of the mesocosm units and the TPC determined from the processed samples (Boxhammer et al., 2016) as described above.

Mesozooplankton was collected by net hauls (100 µm mesh size), fixed (ethanol) and counted in a stereomicroscope. Zooplankton carbon biomass (CB) was calculated using the displacement volume (DV) and the equation of Wiebe (1988): (log DV + 1.429)/0.82 = log CB. Micro and nanoplankton (zoo- and phytoplankton) CB was determined from microscopic counts of fixed (acidic Lugol's iodine solution) samples, and the cellular bio-volumes were determined according to Olenina et al. (2006) and converted to POC by the equations provided by Menden-Deuer and Lessard (2000).

Picophytoplankton were counted using flow cytometry and converted to CB by size fractionation (Veldhuis and Kraay, 2004) and cellular carbon conversion factors (0.2 pg C $µm^{-3}$ (Waterbury et al., 1986). Prokaryotes and viruses were determined according to Marie et al. (1999) and Brussaard (2004), respectively. All heterotrophic prokaryotes, hereafter termed bacteria, and viruses were converted to CB assuming 12.5 fg C $cell^{-1}$ (Heinänen and Kuparinen, 1991) and 0.055 fg C $virus^{-1}$ (Steward et al., 2007), respectively.

The respiration rate was calculated from the difference between the $O_2$ concentration (measured with a Fibox 3, PreSens) before and after a 48 h incubation period in a dark, climate controlled room set to the average temperature observed in the mesocosms.

Bacterial protein production (BPP) was determined by [14]C-leucine ([14]C-Leu) incorporation (Simon and Azam, 1989) according to Grossart et al. (2006). The amount of incorporated [14]C-Leu was converted into BPP by using an intracellular isotope dilution factor of 2. A conversion factor of 0.86 was used to convert the produced protein into carbon (Simon and Azam, 1989).

Net primary production (NPP) was measured using radio-labeled $NaH^{14}CO_3$ (Steeman-Nielsen, 1952). Samples were incubated for 24 h in duplicate, 8 ml vials moored on small incubation platforms at 2, 4, 6, 8 and 10 m depth next to the mesocosms. The areal primary production was calculated based on a simple linear model of the production measurements from the different depths (Spilling et al., 2016).

## 2.3. Gas exchange

In order to calculate the $CO_2$ gas exchange with the atmosphere ($CO_{2flux}$), we used $N_2O$ as tracer gas, and this was added to mesocosm M5 and M8 (control and high $CO_2$ treatment) according to Czerny et al. (2013b). The $N_2O$ concentration was determined every second day using gas chromatography. Using the $N_2O$ measurements, the fluxes across the water surface ($F_{N2O}$) was calculated according to:

$$F_{N2O} = I_{t1} - I_{t2} / (A * \Delta t) \tag{1}$$

where $I_{t1}$ and $I_{t2}$ is the bulk $N_2O$ concentration at time: $t_1$ and $t_2$; A is the surface area and $\Delta t$ is the time difference between $t_1$ and $t_2$.

The flux velocity was then calculated by:

$$K_{N2O} = F_{N2O} / (C_{N2Ow} - (C_{N2O\ aw})) \tag{2}$$

where $C_{N2Ow}$ is the bulk $N_2O$ concentration in the water at a given time point, and $C_{N2Oaw}$ is the equilibrium concentration for $N_2O$ (Weiss and Price, 1980).

The flux velocity for $CO_2$ was calculated from the flux velocity of $N_2O$ according to:

$$k_{CO2} = k_{N2O} / (Sc_{CO2}/Sc_{N2O})^{0.5} \tag{3}$$

where $Sc_{CO2}$ and $Sc_{N2O}$ are the Schmidt numbers for $CO_2$ and $N_2O$, respectively. The $CO_2$flux across the water surface was calculated according to:

$F_{CO2} = k_{CO2} (C_{CO2w} - C_{CO2aw})$                    (4)
where $C_{CO2w}$ is the water concentration of $CO_2$ and $C_{CO2aw}$ is the equilibrium concentration of
$CO_2$. $CO_2$ is preferentially taken up by phytoplankton at the surface, where also the
atmospheric exchange takes place. For this reason, we used the calculated $CO_2$ concentration
(based on the integrated $CO_2$ concentration and pH in the surface) from the upper 5 m as the
input for equation 5.
In contrast to $N_2O$, the $CO_2$ flux can be chemically enhanced by hydration reactions of $CO_2$
with hydroxide ions and water molecules in the boundary layer (Wanninkhof and Knox,
1996). Using the method outlined in Czerny et al. (2013b) we found an enhancement of up to
12% on warm days and this was included into our flux calculations.

**2.4.    Data treatment**
The primary data generated in this study comprise of carbon standing stock measurements of
TPC, DOC, DIC, as well as carbon estimates of meso- and microzooplankton, micro-, nano-
and picophytoplankton, bacteria and viruses. Flux measurements of atmospheric $CO_2$
exchange and sedimentation of TPC, as well as the biological rates of net primary production
($NPP_{14C}$), bacterial production (BP) and total respiration (TR) enabled us to make carbon
budget.
Based on the primary variables (Chl *a* and temperature), the experiment where divided into
three distinct phases: Phase I: *t0-t16*; Phase II: *t17-t30* and Phase III: *t31-t43,* where e.g.
Chlorophyll *a* (Chl *a*) concentration was relatively high during Phase I, decreased during
Phase II and remained low during Phase III (Paul et al. 2015). Measurements of pools and
rates were average for the two first sampling points of each experimental phase (n = 2) and
where normalized to $m^2$ knowing the total depth (17 m, excluding the sedimentation funnel)
of the mesocosms. For Phase III we used the average of the last two measurements as the end
point (n = 2).
For fluxes and biological rates we used the average for the whole periods normalized to days
($day^{-1}$),. The same was done for rates of change ($\Delta$TPC, $\Delta$DOC and $\Delta$DIC), which accounted
for the difference between the start and end of each phase for all carbon pools ($TPC_{pool}$,
$DOC_{pool}$, $DIC_{pool}$). All error estimates were calculated as standard error (SE), and this was
calculated using all measurements within each phase (e.g. calculating the $\Delta$TPC SE using the

difference between each TPC measurement). The three different phases of the experiments were of different length and each variable had a slightly different sampling regime (every 1-3 days, and some measurements missing due to technical problems). The exact sample number (n) for each SE is presented in the Table legends 1-3. The SE for estimated rates were calculated from the square root of the sum of variance for all the variables (Eq 5-10 below) The primary papers mentioned above (section 2.2.) present detailed statistical analyses and we only refer to those here.

NPP was measured directly and we additionally estimated the net community production (NCP). This was done in two different ways from the organic ($NCP_o$), dissolved plus particulate and inorganic ($NCP_i$) fractions of carbon. $NCP_o$ was calculated from changes in the organic fraction plus the exported TPC ($EXP_{TPC}$) according to:

$$NCP_o = EXP_{TPC} + \Delta TPC + \Delta DOC \qquad (5)$$

Direct measurements using $^{14}C$ isotope incubations should in principal provide a higher value than summing up the difference in overall carbon balance (our $NCP_o$), as the latter would incorporate total respiration and not only autotrophic respiration. $NCP_i$ was calculated through changes in the dissolved inorganic carbon pool, corrected for $CO_2$ gas exchange with the atmosphere (CO2flux) according to:

$$NCP_i = CO_{2flux} - \Delta DIC \qquad (6)$$

In order to close the budget we estimated gross primary production (GPP) and DOC production ($DOC_{prod}$). GPP is defined as the photosynthetically fixed carbon without any loss processes (i.e. NPP + autotrophic respiration). GPP can be estimated based on changes in organic ($GPP_o$) or inorganic ($GPP_i$) carbon pools, and we used these two different approaches providing a GPP range:

$$GPP_o = NCP_o + TR \qquad (7)$$

$$GPP_i = TR + CO_{2flux} - \Delta DIC \qquad (8)$$

During Phase III, TR was not measured and we estimated TR based on the ratios between $NCP_o$ and BP to TR during Phase II. The minimum production of DOC ($DOC_{minp}$) in the system was calculated assuming bacterial carbon uptake was taken from the DOC pool according to:

$DOC_{minp} = \Delta DOC + BP$                          (9)
However, this could underestimate $DOC_{prod}$ as a fraction of bacterial DOC uptake is respired.
Without direct measurement of (heterotrophic prokaryote) bacterial respiration, (BR), we
estimated BR from TR. The share of active bacteria contributing to bacterial production is
typically in the range of 10-30% of the total bacterial community (Lignell et al., 2013). We
used the fraction of bacterial biomass (BB) of total biomass (TB) as the maximum limit of
BR (BR $\leq$ BB/TB), and hence calculated max DOC production ($DOC_{maxp}$) according to:
$DOC_{maxp} = \Delta DOC + BP + (BB * TR / TB)$             (10)
We assumed that carbon synthesized by bacteria added to the TPC pool.
There are a number of uncertainties in these calculations, but this budgeting exercise provides
an order-of-magnitude estimate of the flow of carbon within the system and enables
comparison between the treatments. The average of the two controls (M1 and M5) and two
highest $CO_2$ treatments (M3 and M8) were used to illustrate $CO_2$ effects.

**3. Results and discussion**
**3.1 Change in plankton community, from large to small forms over time**
The overall size structure of the plankton community decreased over the course of the
experiment. Fig 1 illustrates the carbon content in different plankton groups in the control
mesocosms. During Phase I, the phytoplankton abundances increased at first in all treatments
before starting to decrease at the end of Phase I (Paul et al., 2015). At the start of Phase II
(t17), the phytoplankton biomass was higher than at the start of the experiment (~130 mmol
C m$^{-2}$ in the controls) but decreased throughout Phase II and III. The fraction of
picophytoplankton increased in all treatments, but some groups of picophytoplankton
increased more in the high $CO_2$ treatments (Crawfurd et al., 2016).
Nitrogen was the limiting nutrient throughout the entire experiment (Paul et al., 2015), and
primary producers are generally N-limited in the main sub-basins of the Baltic Sea
(Tamminen and Andersen, 2007). The surface to volume ratio increases with decreasing cell
size, and consequently small cells have higher nutrient affinity, and are better competitors for
scarce nutrient sources than large cells (Reynolds, 2006). The prevailing N-limitation was
likely the reason for the decreasing size structure of the phytoplankton community.

Micro and mesozooplankton standing stock was approximately half of the phytoplankton biomass initially, but decreased rapidly in the control treatments during Phase I (Fig 1). In the $CO_2$ enriched treatments the zooplankton biomass also decreased but not to the same extent as in the control treatments (Spilling et al., 2016). Overall, smaller species benefitted from the extra $CO_2$ addition, but there was no significant negative effect of high $CO_2$ on the mesozooplankton community (Lischka et al., 2015).

Bacterial biomass was the main fraction of the plankton carbon throughout the experiment. The bacterial numbers largely followed the phytoplankton biomass with an initial increase then decrease during Phase I; increase during Phase II and slight decrease during Phase III (Crawfurd et al., 2016). The bacterial community was controlled by mineral nutrient limitation, bacterial grazing and viral lysis (Crawfurd et al., 2016), and bacterial growth is typically limited by N or a combination of N and C in the study area (Lignell et al., 2008; Lignell et al., 2013).

The bacterial carbon pool was higher than the measured TPC. Part of the bacteria must have passed the GFF filters (0.7 µm), and assuming pico- to mesoplankton was part of the TPC, >50% of the bacterial carbon was not contributing to the measured TPC. The conversion factor from cells to carbon is positively correlated to cell size, and there is consequently uncertainty related to the absolute carbon content of the bacterial pool (we used a constant conversion factor). However, bacteria is known to be the dominating carbon share in the Baltic Sea during the N-limited summer months (Lignell et al., 2013), and its relative dominance is in line with this.

Although there are some uncertainty in the carbon estimate (Jover et al. 2014), virus make up (due to their numerical dominance) a significant fraction of the pelagic carbon pool. Of the different plankton fractions the virioplankton have been the least studied, but their role in the pelagic ecosystem is ecologically important (Suttle, 2007; Brussaard et al., 2008; Mojica et al., 2016). Viral lysis rates were equivalent to the grazing rates for phytoplankton and for bacteria in the current study (Crawfurd et al., 2015). As mortality agents, viruses are key drivers of the regenerative microbial food web (Suttle, 2007; Brussaard et al., 2008). Overall, the structure of the plankton community reflected the nutrient status of the system. The increasing N-limitation favoring development of smaller cells, and increasing dependence of the primary producers on regenerated nutrients.

**3.2. The DIC pool and atmospheric exchange of $CO_2$**

The DIC pool was the largest carbon pool: 3-4 fold higher than the DOC pool and roughly 60-fold higher than the TPC pool (Tables 1-3). After the addition of $CO_2$, the DIC pool was ~7% higher in the highest $CO_2$ treatment compared to the control mesocosms (Table 1). The gas exchange with the atmosphere was the most apparent flux affected by $CO_2$ addition (Tables 1-3). Seawater in the mesocosms with added $CO_2$ were supersaturated, hence $CO_2$ outgassed throughout the experiment. The control mesocosms were initially undersaturated, hence ingassing occurred during Phases I and II (Fig 2). In the first part of Phase III, the control mesocosms reached equilibrium with the atmospheric $fCO_2$ (Fig. 2). The gas exchange had direct effects on the DIC concentration in the mesocosms (Fig. 3). From the measured gas exchange and change in DIC it is possible to calculate the biologically mediated carbon flux. In the mesocosms with ambient $CO_2$ concentration, the flux measurements indicated net heterotrophy throughout the experiment. The opposite pattern, net autotrophy, was indicated in the two mesocosms with the highest $CO_2$ addition (Fig 3; see also section 3.7.).

**3.3. The DOC pool, DOC production and remineralization**

The DOC pool increased throughout the experiment in all mesocosm bags, but more in the treatments with elevated $CO_2$ concentration. The initial DOC standing stock in all treatments was approximately 7200 mmol C m$^{-2}$. At the end of the experiment, the DOC pool was ~2% higher in the two highest $CO_2$ treatments compared to the controls (Fig. 4), and there is statistical support for this difference between $CO_2$ treatments (Phase III, $p = 0.05$) (Paul et al., 2015). Interestingly, the data does not point to a substantially higher release of DOC at high $CO_2$ (Figs 4 and 5). The bacterial production was notably lower during Phases II and III in the high $CO_2$ treatments (Hornick et al., 2016), and of similar magnitude as the rate of change in DOC pool (Table 2 and 3), indicating reduced bacterial uptake and remineralization of DOC. The combined results suggest that the increase in the DOC pool at high $CO_2$ was related to reduced DOC loss (uptake by bacteria), rather than increased release of DOC by the plankton community, at elevated $CO_2$ concentration.

The Baltic Sea is affected by large inflow of freshwater containing high concentrations of refractory DOC such as humic substances, and the concentration in Gulf of Finland is typically 400-500 $\mu$mol C L$^{-1}$ (Hoikkala et al., 2015). The large pool of DOC and turn over

times of ~200 days (Tables 1-3) is most likely a reflection of the relatively low fraction of
labile DOC, but bacterial limitation of mineral nutrients can also increase turn over times
(Thingstad et al., 1997).
The DOC pool has been demonstrated to aggregate into transparent exopolymeric particles
(TEP) under certain circumstances, which can increase sedimentation at high $CO_2$ levels
(Riebesell et al., 2007). We did not have any direct measurements of TEP, but any $CO_2$ effect
on its formation is highly dependent on the plankton community and its physiological status
(MacGilchrist et al., 2014). No observed effect of $CO_2$ treatment on carbon export suggests
that we did not have a community where the TEP production was any different between the
treatments used.

### 3.4. The TPC pool and export of carbon

There was a positive effect of elevated $CO_2$ on TPC relative to the controls. At the start of the
experiment, the measured TPC concentration in the enclosed water columns was 400-500
mmol C m$^{-2}$ (Table 1). The TPC pool decreased over time but less in the high $CO_2$ treatment
and at the end of the experiment, the standing stock of TPC was ~6% higher (Phase III, p =
0.01; Paul et al. (2015) in the high $CO_2$ treatment (Fig. 4).
The export of TPC was not dependent on the $CO_2$ concentration but varied temporally. The
largest flux of TPC out of the mesocosms occurred during Phase I with ~6 mmol C m$^{-2}$ d$^{-1}$. It
decreased to ~3 mmol C m$^{-2}$ d$^{-1}$ during Phase II and was ~2 mmol C m$^{-2}$ d$^{-1}$ during Phase III
(Table 1-3). The exported carbon as percent of average TPC standing stock similarly
decreased from ~1.3% during Phase I to 0.3-0.5% during Phase III. The initial increase in the
autotrophic biomass was the likely reason for relatively more of the carbon settling in the
mesocosms in the beginning of the experiment whereas the decreasing carbon export was
most likely caused by the shift towards a plankton community depending on recycled
nitrogen. This reduced the overall suspended TPC and also the average plankton size in the
community.

### 3.5. Biological rates: respiration

Total respiration (TR) was always lower in the $CO_2$ enriched treatments (Tables 1-3). The
average TR was 83 mmol C m$^{-2}$ d$^{-1}$ during Phase I, and initially without any detectable
treatment effect. The respiration rate started to be lower in the high $CO_2$ treatments,
compared with the controls, in the beginning of Phase II. At the end of Phase II there was a
significant difference (p = 0.02; Spilling et al., 2016) between the treatments (Table 2), and
40% lower respiration rate in the highest $CO_2$ treatment compared with the controls (Spilling
et al., 2016).
Cytosol pH is close to neutral in most organisms, and reduced energetic cost for internal pH
regulation (e.g. transport of $H^+$) and at lower external pH levels could be one factor reducing
respiration (Smith and Raven, 1979). Hopkinson et al. (2010) found indirect evidence for
decreased respiration and also proposed that increased $CO_2$ concentration (i.e. decreased pH)
reduced metabolic cost of remaining intracellular homeostasis. Mitochondrial respiration in
plant foliage decreases in high $CO_2$ environments, possibly affected by respiratory enzymes
or other metabolic processes (Amthor, 1991; Puhe and Ulrich, 2012). Most inorganic carbon
in water is in the form of bicarbonate ($HCO_3^-$) at relevant pH, and many aquatic autotrophs
have developed carbon concentrating mechanisms (CCMs) (e.g. Singh et al., 2014) that could
reduce the cost of growth (Raven, 1991). There are some studies that have pointed to savings
of metabolic energy due to down-regulation of carbon concentrating mechanisms (Hopkinson
et al., 2010) or overall photosynthetic apparatus (Sobrino et al., 2014) in phytoplankton at
high $CO_2$ concentrations. Yet, other studies of the total plankton community have pointed at
no effect or increased respiration at elevated $CO_2$ concentration (Li and Gao, 2012; Tanaka et
al., 2013), and the metabolic changes behind reduced respiration, remains an open question.
Membrane transport of H+ is sensitive to changes in external pH, but the physiological
impacts of increasing H+ needs further study to better address effects of ocean acidification
(Taylor et al., 2012). An important aspect is also to consider the microenvironment
surrounding plankton; exchange of nutrients and gases takes place through the boundary
layer, which might have very different pH properties than bulk water measurements (Flynn et
al., 2012).

**3.6. Biological rates: bacterial production**
Bacterial production (BP) became lower in the high $CO_2$ treatment in the latter part of the
experiment. During Phase I, BP ranged from 27 to 46 mmol C $m^{-2}$ $d^{-1}$ (Table 1). The
difference in BP between treatments became apparent in Phases II and III of the experiment.
The average BP was 18% and 24% higher in the controls compared to the highest $CO_2$
treatments during Phases II and III, respectively (Tables 2 and 3). Statistical support (p≤0.01)
for a treatment effect during parts of the experiment is presented in Hornick et al. (2016).
The lower bacterial production accounted for ~40% of the reduced respiration during Phase
II, and the reduced respiration described above could at least partly be explained by the lower
bacterial activity. This raises an interesting question: what was the mechanism behind the
reduced bacterial production/respiration in the high $CO_2$ treatment? There are examples of
decreased bacterial production (Motegi et al 2013) and respiration (Teira et al., 2012) at
elevated $CO_2$ concentration. However, most previous studies have reported no change
(Allgaier et al., 2008) or a higher bacterial production at elevated $CO_2$ concentration
(Grossart et al., 2006; Piontek et al., 2010; Endres et al., 2014). The latter was also supported
by the recent study of Bunse et al. (2016), describing up-regulation of bacterial genes related
to respiration, membrane transport and protein metabolism at elevated $CO_2$ concentration;
albeit, this effect was not evident when inorganic nutrients had been added (high Chl *a*
treatment).
In this study, the reason for the lower bacterial activity in the high $CO_2$ treatments could be
due to either limitation and/or inhibition of bacterial growth or driven by difference in loss
processes. Bacterial grazing and viral lysis was higher in the high $CO_2$ treatments during
periods of the experiment (Crawfurd et al., 2016), and would at least partly be the reason for
the reduced bacterial production at high $CO_2$ concentration.
N-limitation increased during the experiment (Paul et al., 2015), and mineral nutrient
limitation of bacteria can lead to accumulation of DOC, i.e. reduced bacterial uptake
(Thingstad et al., 1997), similar to our results. Bacterial N limitation is common in the area
during summer (Lignell et al., 2013), however, this N-limitation was not apparently different
in the controls (Paul et al., 2015), and $CO_2$ did not affect N-fixation (Paul et al., 2016). In a
scenario where the competition for N is fierce, the balance between bacteria and similar sized
picophytoplankton could be tilted in favor of phytoplankton if they gain an advantage by
having easier access to carbon, i.e. $CO_2$ (Hornick et al., 2016). We have not found evidence
in the literature that bacterial production will be suppressed in the observed pH range inside
the mesocosms, varying from approximately pH 8.1 in the control to pH 7.6 in the highest
$f$$CO_2$ treatment (Paul et al., 2015), although enzyme activity seems to be affected even by
moderate pH changes. For example, some studies report on an increase in protein degrading
enzyme leucine aminopeptidase activities at reduced pH (Grossart et al., 2006; Piontek et al.,
2010; Endres et al., 2014), whereas others indicate a reduced activity of this enzyme
(Yamada and Suzumura, 2010). A range of other factors affects this enzyme, for example the
nitrogen source and salinity (Stepanauskas et al., 1999), and any potential interaction effects
with decreasing pH are not yet resolved. Any pH-induced changes in bacterial enzymatic
activity could potentially affect bacterial production.

**3.7. Biological rates: primary production**
There was an effect of $CO_2$ concentration on the net community production based on the
organic carbon fraction ($NCP_o$). $NCP_o$ was higher during Phase I than during the rest of the
experiments and during this initial phase without any apparent $CO_2$ effect. There was no
consistent difference between $CO_2$ treatments for $NPP_{14C}$ ($p > 0.1$), but $NCP_o$ increased with
increasing $CO_2$ enrichment during Phase II (Phase II; linear regression $p = 0.003$; $R^2 = 0.91$).
This was caused by the different development in the TPC and DOC pools. The pattern of
gross primary production (GPP) was similar to $NCP_o$ during Phases I and II. During Phase III
there were no respiration or $NPP_{14C}$ measurements and the estimated GPP is more uncertain.
The $NCP_o$ and GPP indicated a smaller difference between treatments during Phase III
compared with Phase II.
The measures of $NPP_{14C}$ and $NCP_o$ were of a similar magnitude (Tables 1-3). During Phase I,
$NPP_{14C} < NCP_o$ (Table 1), this relationship reversed for most treatments during Phase II, with
the exception of the highest $CO_2$ levels (Table 2). The difference between $NPP_{14C}$ and $NCP_o$
suggests that observed reduction in respiration at elevated $CO_2$ could be mainly heterotrophic
respiration. However, in terms of the $NPP_{14C} < NCP_o$, the uncertainty seems to be higher than
the potential signal of heterotrophic respiration. This would also indicate that the $NPP_{14C}$
during Phase I have been underestimated, in particular for the control mesocosm M1. During
Phase II, the $NPP_{14C}$ was higher than $NCP_o$, except for the two highest $CO_2$ treatments, more
in line with our assumption of $NPP_{14C} > NCP_o$. The systematic offset in $NPP_{14C}$ during Phase
I could be due to changed parameterization during incubation in small volumes (8 mL,
Spilling et al., 2016), for example increased loss due to grazing.
The results of the DIC pool and atmospheric exchange of $CO_2$ provides another way of
estimating the net community production based on inorganic carbon ($NCP_i$). There was some
discrepancy between the $NCP_o$ and $NCP_i$ as the latter suggested net heterotrophy in the
ambient $CO_2$ whereas the high $CO_2$ treatments were net autotrophic during all three phases of
the experiment (Fig. 3). For the NCPo there was no indication of net heterotrophy at ambient
$CO_2$ concentration. In terms of the absolute numbers, the NCPi estimate is probably more
uncertain than $NCP_o$. Calculating the $CO_2$ atmospheric exchange from the measurements of a
tracer gas involves several calculation steps (Eq 1-4), each adding uncertainty to the
calculation. However, both estimations (NCPi and NCPo) indicate that increased $CO_2$
concentrations lead to higher overall community production, supporting our overall
conclusion.


**3.8 Budget**
A carbon budget for the two control mesocosms and two highest $CO_2$ additions is presented
in Fig. 5. During Phase I the estimated gross primary production (GPP) was ~100 mmol C
fixed $m^{-2}$ $d^{-1}$; from which 75-95% were respired, ~1% ended up in the TPC (including export)
and 5-25% added to the DOC pool. The main difference between $CO_2$ treatments became
apparent during Phase II when the $NCP_o$ was higher in the elevated $CO_2$ treatments. The
respiration loss increased to ~100% of GPP at the ambient $CO_2$ concentration, whereas
respiration was lower (85-95% of GPP) in the highest $CO_2$ treatment. Bacterial production
was ~30% lower, on average, at the highest $CO_2$ concentration compared with the controls
during Phase II. The share of $NCP_o$ of GPP ranged from 2% to 20% and the minimum flux to
the DOC pool was 11% to 18% of TPC.
The overall budget was calculated by using the direct measurements of changes in standing
stocks and fluxes of export, respiration and bacterial production rates. The most robust data
are the direct measurements of carbon standing stocks and their development (e.g. ΔTPC).
These are based on well-established analytical methods with relatively low standard error
(SE) of the carbon pools. However, the dynamic nature of these pools made the relative SE
for the rate of change much higher, reflecting that the rate of change varied considerably
within the different phases.
The rate variables, calculated based on conversion factors, have greater uncertainty, although
their SEs were relatively low, caused by uncertainty in the conversion steps. For example, the
respiratory quotient (RQ) was set to one, which is a good estimate for carbohydrate oxidation.
For lipids and proteins the RQ is close to 0.7, but in a natural environment RQ is often >1
(Berggren et al., 2012), and is affected by physiological state e.g. nutrient limitation
(Romero-Kutzner et al., 2015). Any temporal variability in the conversion factors would
directly change the overall budget calculations, e.g. RQ affecting total respiration and gross
primary production estimates. However, the budget provides an order-of-magnitude estimate
of the carbon flow within the system. Some of the variables such as GPP were estimated
using different approaches, providing a more robust comparison of the different treatments.
The primary effect of increasing $CO_2$ concentration was the higher standing stocks of TPC
and DOC compared with ambient $CO_2$ concentration. The increasing DOC pool and
relatively higher TPC pool were driven by reduced respiration and bacterial production at
elevated $CO_2$ concentration. Decreasing respiration rate reduced the recycling of organic
carbon back to the DIC pool. The lower respiration and bacterial production also indicates
reduced remineralization of DOC. These two effects caused the higher TPC and DOC pools
in the elevated $CO_2$ treatments. The results highlight the importance of looking beyond net
changes in carbon standing stocks to understand how carbon fluxes are affected under
increasing ocean acidification.


**Acknowledgements**
We would like to thank all of the staff at Tvärminne Zoological station, for great help during
this experiment, and Michael Sswat for carrying out the TPC filtrations. We also gratefully
acknowledge the captain and crew of R/V ALKOR (AL394 and AL397) for their work
transporting, deploying and recovering the mesocosms. The collaborative mesocosm
campaign was funded by BMBF projects BIOACID II (FKZ 03F06550) and SOPRAN Phase
II (FKZ 03F0611). Additional financial support for this study came from Academy of Finland
(KS - Decisions no: 259164 and 263862) and Walter and Andrée de Nottbeck Foundation
(KS). TH and HPG were financially supported by SAW project TemBi of the Leibniz
Foundation. CPDB was financially supported by the Darwin project, the Royal Netherlands
Institute for Sea Research (NIOZ), and the EU project MESOAQUA (grant agreement
number 228224).

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

Table 1. The standing stock of total particular carbon ($TPC_{pool}$), dissolved organic carbon ($DOC_{pool}$) and dissolved inorganic carbon ($DIC_{pool}$) at the start of Phase I in mmol C $m^{-2}$ ± SE (n = 2). The $DOC_{pool}$ was missing some initial measurements and is the average for all mesocosms assuming that the DOC concentration was similar at the onset of the experiment. The net change in TPC (ΔTPC), DOC (ΔDOC) and DIC (ΔDIC) are average changes in the standing stocks during Phase I in mmol C $m^{-2}$ $d^{-1}$ ± SE (n = 8). Flux measurements of atmospheric gas exchange ($CO_{2flux}$) and exported carbon ($EXP_{TPC}$) plus biological rates: total respiration (TR), bacterial (BP) and net primary production ($NPP_{14C}$) and net community production estimated based on organic carbon pools ($NCP_o$) net primary production, are all average for the whole Phase I in mmol C $m^{-2}$ $d^{-1}$ ± SE (n = 13, 9, 16, 7 and 11 for $CO_{2flux}$, $EXP_{TCP}$, TR, BP and $NPP_{14C}$ respectively). SE for NCPo was calculated from the square root of the sum of variance of the three variables used in Eq 6. The $NCP_o$ was calculated from the net change in carbon pools plus carbon export, whereas $NPP_{14C}$ was measured carbon fixation using radiolabeled $^{14}C$ over a 24 h incubation period *in situ*. TR was measured as $O_2$ consumption and for comparison with carbon fixation we used a respiratory quotient (RQ) of 1. $CO_{2flux}$ was only calculated for the period after full addition of CO2 (*t4-t16*). A total budget of carbon fluxes for ambient and high $CO_2$ treatments is presented in Fig 5.

**Phase I (*t0-t16*)**

| **CO$_2$ treatment (µatm *f*CO$_2$)** | **365** | **368** | **497** | **821** | **1007** | **1231** |
|---|---|---|---|---|---|---|
| **Mesocosm number** | **M1** | **M5** | **M7** | **M6** | **M3** | **M8** |
| $TPC_{pool}$ | 417 ± 38 | 425 ± 39 | 472 ± 48 | 458 ± 38 | 431 ± 48 | 446 ± 57 |
| $DOC_{pool}$ | 7172 ± 87 | 7172 ± 87 | 7172 ± 87 | 7172 ± 87 | 7172 ± 87 | 7172 ± 87 |
| $DIC_{pool}$ | 25158 ± 9 | 25182 ± 10 | 25628 ± 8 | 26295 ± 22 | 26637 ± 36 | 26953 ± 48 |
| ΔTPC | -4.6 ± 15 | -5.2 ± 13 | -8.3 ± 13 | -8.2 ± 17 | -7.0 ± 13 | -6.3 ± 20 |
| ΔDOC | 15.5 ± 58 | 18.3 ± 30 | 18.5 ± 33 | 25.0 ± 36 | 18. 5 ± 73 | 18.1 ± 63 |
| ΔDIC | 5.5 ± 5.2 | 6.9 ± 9.2 | -6.1 ± 11 | -24 ± 14 | -32 ± 20 | -49 ± 42 |
| $CO_{2flux}$ | 4.4 ± 0.2 | 4.8 ± 0.3 | -0.8 ± 0.5 | -11 ± 1.0 | -17 ± 1.4 | -23 ± 2.0 |
| $EXP_{TPC}$ | 6.6 ± 0.10 | 5.6 ± 0.04 | 5.4 ± 0.07 | 6.0 ± 0.07 | 5.6 ± 0.06 | 6.0 ± 0.05 |
| TR | 107 ± 9 | 82 ± 7 | 81 ± 6 | 80 ± 8 | 75 ± 8 | 74 ± 8 |
| BP | 27 ± 8 | 41 ± 6 | 43 ± 8 | 41 ± 4 | 36 ± 5 | 46 ± 9 |
| $NPP_{14c}$ | 4.8 ± 0.8 | 11.4 ± 2.1 | 14.9 ± 3.6 | 12.3 ± 2.3 | 11.3 ± 2.4 | 14.5 ± 2.7 |
| $NCP_o$ | 17.4 ± 33 | 18.7 ± 20 | 15.6 ± 30 | 22.8 ± 28 | 17.1 ± 25 | 17.8 ± 28 |

Table 2. The standing stock of total particular carbon (TPC$_{pool}$), dissolved organic carbon (DOC$_{pool}$) and dissolved inorganic carbon (DIC$_{pool}$) at the start of Phase II in mmol C m$^{-2}$ ± SE (n = 2). The net change in TPC (ΔTPC), DOC (ΔDOC) and DIC (ΔDIC) are average changes in the standing stocks during Phase II in mmol C m$^{-2}$ d$^{-1}$ ± SE (n = 7). Flux measurements of atmospheric gas exchange (CO$_{2flux}$) and exported carbon (EXP$_{TPC}$) plus biological rates: total respiration (TR), bacterial production (BP), measured (NPP$_{14C}$) and net community production estimated based on organic carbon pools (NCP$_o$), are all average for Phase II in mmol C m$^{-2}$ d$^{-1}$ ± SE (n = 8, 7, 14, 5 and 14 for CO$_{2flux}$, EXP$_{TCP}$, TR, BP and NPP$_{14C}$ respectively). See Table 1 legend for further details.

**Phase II (*t17-t30*)**

| CO$_2$ treatment (µatm *f*CO$_2$) | 365 | 368 | 497 | 821 | 1007 | 1231 |
|---|---|---|---|---|---|---|
| **Mesocosm number** | M1 | M5 | M7 | M6 | M3 | M8 |
| TPC$_{pool}$ | 339 ± 14 | 337 ± 20 | 331 ± 22 | 318 ± 9 | 312 ± 12 | 339 ± 23 |
| DOC$_{pool}$ | 7435 ± 38 | 7483 ± 37 | 7487 ± 43 | 7597 ± 37 | 7487 ± 61 | 7479 ± 37 |
| DIC$_{pool}$ | 25247 ± 34 | 25269 ± 34 | 25639 ± 8 | 26177 ± 25 | 26413 ± 28 | 26757 ± 45 |
| ΔTPC | -2.4 ± 5 | -2.3 ± 8 | -1.6 ± 14 | 0.3 ± 6 | 2.8 ± 4 | 3.2 ± 8 |
| ΔDOC | -0.6 ± 39 | 2.4 ± 30 | 3.6 ± 40 | 8.4 ± 31 | 11.3 ± 58 | 9.1 ± 36 |
| ΔDIC | 22.4 ± 12 | 17.6 ± 8.1 | -0.4 ± 4.5 | -10.5 ± 16 | -14.2 ± 10 | -23.1 ± 13 |
| CO$_{2flux}$ | 1.7 ± 0.3 | 1.2 ± 0.3 | -2.6 ± 0.3 | -10 ± 0.5 | -14 ± 0.6 | -19 ± 1.0 |
| EXP$_{TPC}$ | 3.3 ± 0.08 | 2.6 ± 0.06 | 2.5 ± 0.08 | 2.6 ± 0.06 | 2.8 ± 0.07 | 2.9 ± 0.06 |
| TR | 140 ± 7 | 127 ± 5 | 103 ± 3 | 103 ± 4 | 101 ± 5 | 86 ± 4 |
| BP | 66 ± 17 | 57 ± 8 | 61 ± 7 | 57 ± 7 | 43 ± 6 | 47 ± 6 |
| NPP$_{14c}$ | 3.8 ± 0.6 | 11.2 ± 1.9 | 10.8 ± 2.0 | 14.3 ± 2.8 | 10.4 ± 2.1 | 12.0 ± 2.5 |
| NCP$_o$ | 0.3 ± 20 | 2.7 ± 15 | 4.5 ± 22 | 11.4 ± 16 | 16.9 ± 19 | 15.2 ± 16 |

Table 3. The standing stock of total particular carbon ($TPC_{pool}$), dissolved organic carbon ($DOC_{pool}$) and dissolved inorganic carbon ($DIC_{pool}$) at the start of Phase III in mmol C $m^{-2}$ ± SE (n = 2). The net change in TPC (ΔTPC), DOC (ΔDOC) and DIC (ΔDIC) are average changes in the standing stocks during Phase III in mmol C $m^{-2}$ $d^{-1}$ ± SE (n = 6), using the average of the last two sampling days as the end point. Flux measurements of atmospheric gas exchange ($CO_{2flux}$) and exported carbon ($EXP_{TPC}$) plus biological rates: bacterial production (BP) and net community production estimated based on organic carbon pools ($NCP_o$), are all average for Phase III in mmol C $m^{-2}$ $d^{-1}$ ± SE (n = 7, 6, and 7 for $CO_{2flux}$, $EXP_{TCP}$, and BP respectively). See Table 1 legend for further details. During Phase III we did not have direct measurements of net primary production ($NPP_{14C}$) or total respiration (TR).

**Phase III (*t31-t43*)**

| CO₂ treatment (µatm *f*CO₂) | 365 | 368 | 497 | 821 | 1007 | 1231 |
|---|---|---|---|---|---|---|
| **Mesocosm number** | **M1** | **M5** | **M7** | **M6** | **M3** | **M8** |
| $TPC_{pool}$ | 306 ± 12 | 304 ± 20 | 309 ± 20 | 323 ± 2 | 351 ± 13 | 384 ± 16 |
| $DOC_{pool}$ | 7426 ± 16 | 7469 ± 20 | 7485 ± 92 | 7553 ± 20 | 7593 ± 30 | 7562 ± 38 |
| $DIC_{pool}$ | 25557 ± 9 | 25545 ± 10 | 25648 ± 13 | 26030 ± 19 | 26197 ± 31 | 26371 ± 32 |
| ΔTPC | -3.8 ± 10 | 0.3 ± 7 | 3.3 ± 14 | 3.3 ± 10 | -1.4 ± 8 | -4.8 ± 8 |
| ΔDOC | 9.8 ± 5 | 8.8 ± 7 | 8.9 ± 43 | 9.2 ± 10 | 5.7 ± 17 | 16.3 ± 20 |
| ΔDIC | 4.3 ± 3.9 | 5.5 ± 8.7 | 6.2 ± 11 | -12.3 ± 7.2 | -16.3 ± 14 | -20.1 ± 14 |
| $CO_{2flux}$ | -0.3 ± 0.7 | -0.8 ± 0.6 | -3.0 ± 0.5 | -7.3 ± 0.5 | -9.4 ± 0.6 | -13 ± 0.6 |
| $EXP_{TPC}$ | 1.5 ± 0.07 | 1.4 ± 0.05 | 0.4 ± 0.07 | 1.9 ± 0.05 | 1.6 ± 0.04 | 1.7 ± 0.05 |
| BP | 31 ± 6.8 | 37 ± 1.4 | 38 ± 1.4 | 27 ± 2.1 | 17 ± 3.8 | 28 ± 2.3 |
| $NCP_o$ | 7.6 ± 16 | 10.5 ± 13 | 12.7 ± 20 | 14.3 ± 13 | 6.0 ± 10 | 13.2 ± 14 |

**Figure legends**

Fig. 1. The different fractions of carbon in the control mesocosms (M1 and M5) at the start of Phase I (t0), II (t17) and III (t31) in mmol C $m^{-2}$ ± SE (n = 2). The differences between the controls and elevated $CO_2$ concentration are discussed in the text. The size of the boxes indicates the relative size of the carbon standing stocks.

Fig 2. The calculated exchange of $CO_2$ between the mesocosms and the atmosphere. Positive values indicate net influx (ingassing) and negative values net outflux (outgassing) from the mesocosms. The flux was based on measurements of $N_2O$ as a tracer gas and calculated using equations 2-5.

Fig 3. Change in dissolved inorganic carbon (DIC) pool and the atmospheric $CO_2$ exchange (Fig. 2). All values are average mmol C $m^{-2}$ $d^{-1}$ ± SE for the three different phases (n = 13, 8 and 7 for Phases I – III respectively) in the control mesocosms (M1 + M5) and high $CO_2$ mesocosms (M3 + M8). Black, solid arrows indicated measured fluxes. Grey, dashed arrows are estimated by closing the budget, and indicate the net community production based on inorganic carbon budget ($NCP_i$), which equals biological uptake or release of $CO_2$.

Fig 4. Standing stocks of total particulate carbon (TPC) and dissolved carbon (DOC) at the last day of the experiment (*t43*), plus the sum of exported TPC throughout the experiment; all values are in mmol C $m^{-2}$ ± SE (n = 2). The values are averages of the two controls (M1 and M5) and the two highest $CO_2$ treatments (M3 and M8). Red circles indicate statistically significant higher standing stocks in the high $CO_2$ treatments (further details in text). The size of the boxes indicates the relative size of the carbon standing stocks and export.

Fig 5. Average carbon standing stocks and flow in the control mesocosms (M1 + M5) and high $CO_2$ mesocosms (M3 + M8) during the three phases of the experiment. All carbon stocks (squares): dissolved inorganic carbon (DIC), total particulate carbon (TPC) and dissolved organic carbon (DOC), are average from the start of the period in mmol C $m^{-2}$ ± SE (n = 2). Fluxes (arrows) and net changes (Δ) are averages for the whole phase in mmol C $m^{-2}$ $d^{-1}$ ± SE (n presented in Table legends 1-3) . Black, solid arrows indicated measured fluxes (Tables 1-3): total respiration (TR), bacterial production (BP), exported TPC ($EXP_{TPC}$). Grey, dashed arrows are estimated by closing the budget: gross primary production (GPP) using equations 7 and 8; DOC production ($DOC_{prod}$) using equations 9 and 10. Bacterial respiration

was calculated using equation 10 and is a share of TR (indicated by the parenthesis).
Aggregation was assumed to equal BP. Red circles indicate statistically higher values
compared with the other $CO_2$ treatment ($p < 0.05$, tests presented in the primary papers
described in section 2.2.). The size of the boxes indicates the relative size of the carbon
standing stocks.

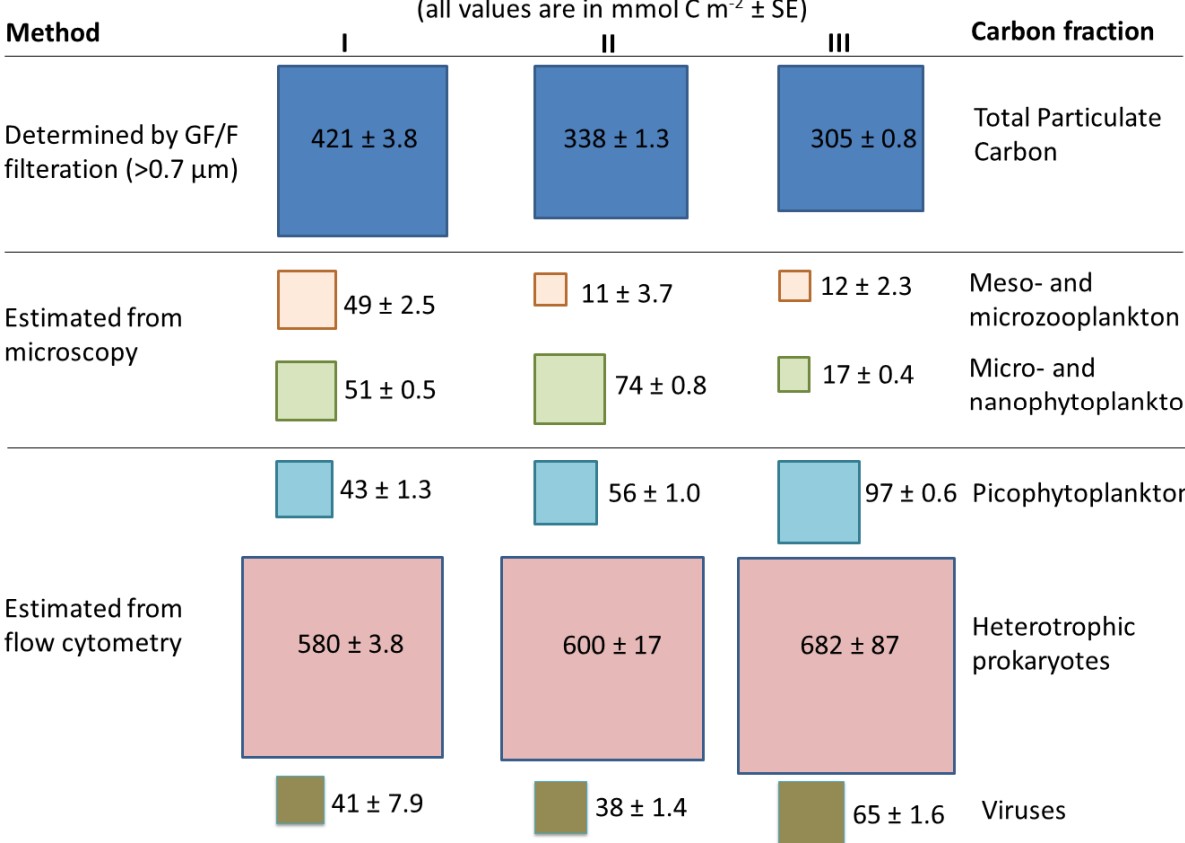

**Experimental phases**
(all values are in mmol C m$^{-2}$ ± SE)

| Method | I | II | III | Carbon fraction |
|---|---|---|---|---|
| Determined by GF/F filteration (>0.7 μm) | 421 ± 3.8 | 338 ± 1.3 | 305 ± 0.8 | Total Particulate Carbon |
| Estimated from microscopy | 49 ± 2.5 | 11 ± 3.7 | 12 ± 2.3 | Meso- and microzooplankton |
| | 51 ± 0.5 | 74 ± 0.8 | 17 ± 0.4 | Micro- and nanophytoplankton |
| | 43 ± 1.3 | 56 ± 1.0 | 97 ± 0.6 | Picophytoplankton |
| Estimated from flow cytometry | 580 ± 3.8 | 600 ± 17 | 682 ± 87 | Heterotrophic prokaryotes |
| | 41 ± 7.9 | 38 ± 1.4 | 65 ± 1.6 | Viruses |

2    **Fig 1**

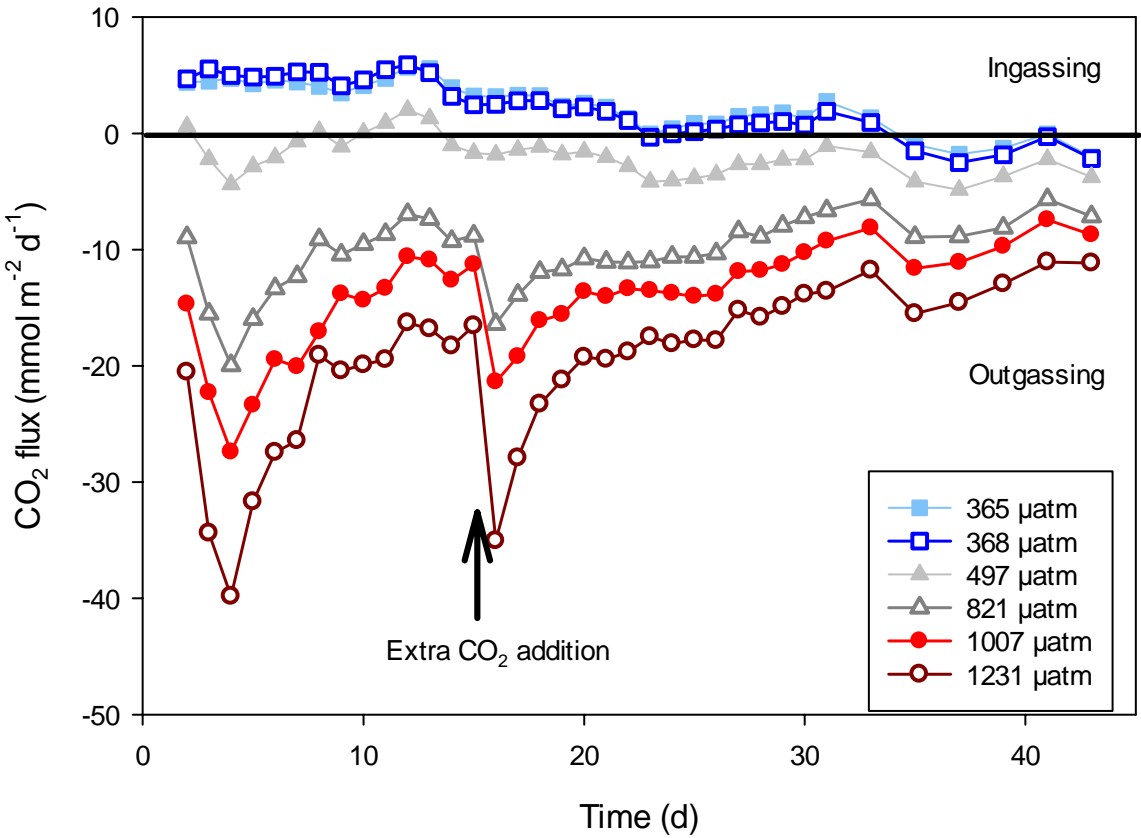

2  **Fig 2**

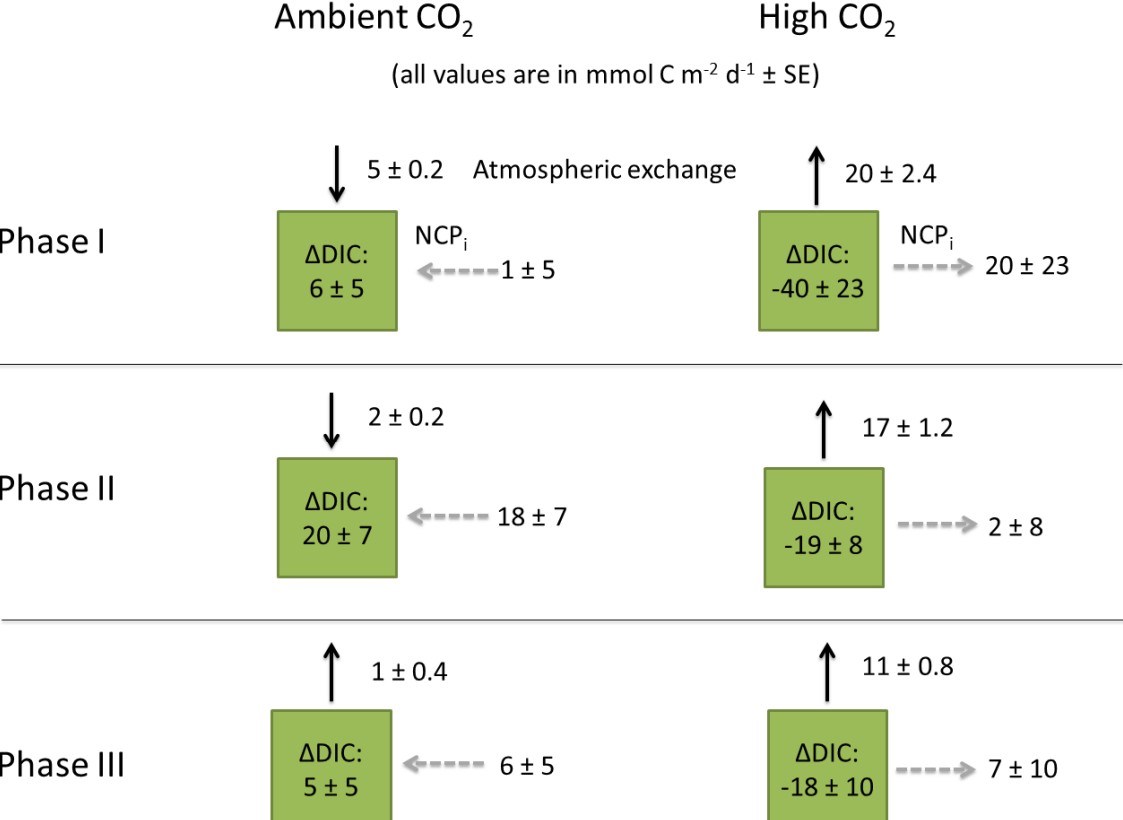

Ambient CO$_2$             High CO$_2$

(all values are in mmol C m$^{-2}$ d$^{-1}$ ± SE)

**Phase I**

5 ± 0.2    Atmospheric exchange      20 ± 2.4

ΔDIC: 6 ± 5    NCP$_i$   1 ± 5

ΔDIC: -40 ± 23    NCP$_i$   20 ± 23

**Phase II**

2 ± 0.2      17 ± 1.2

ΔDIC: 20 ± 7    18 ± 7

ΔDIC: -19 ± 8    2 ± 8

**Phase III**

1 ± 0.4      11 ± 0.8

ΔDIC: 5 ± 5    6 ± 5

ΔDIC: -18 ± 10    7 ± 10

4   **Fig 3**

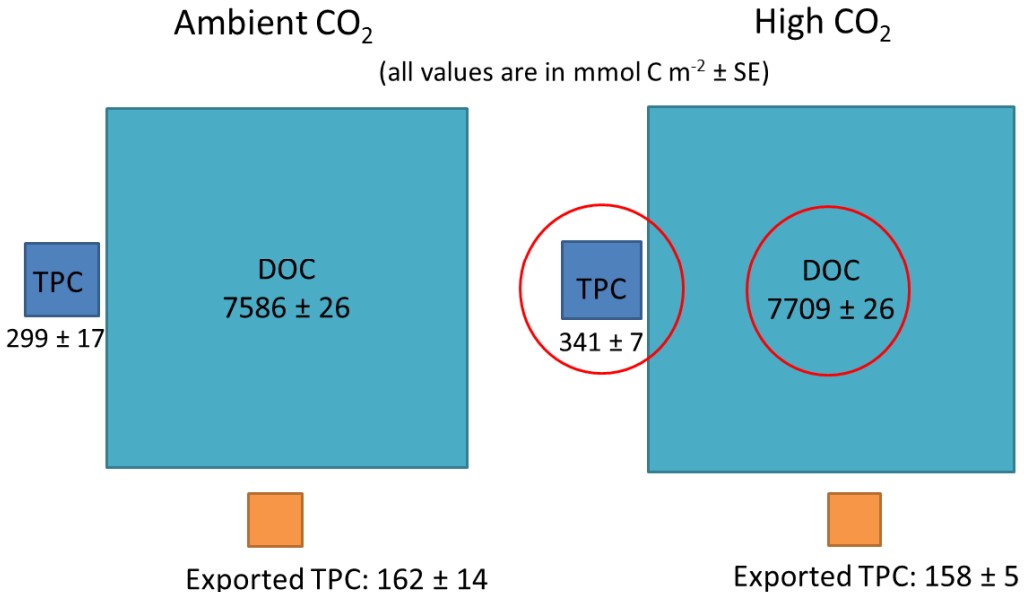

Ambient CO$_2$  ·  High CO$_2$

(all values are in mmol C m$^{-2}$ ± SE)

TPC  ·  DOC 7586 ± 26

299 ± 17

TPC  ·  DOC 7709 ± 26

341 ± 7

Exported TPC: 162 ± 14  ·  Exported TPC: 158 ± 5

3 **Fig 4**

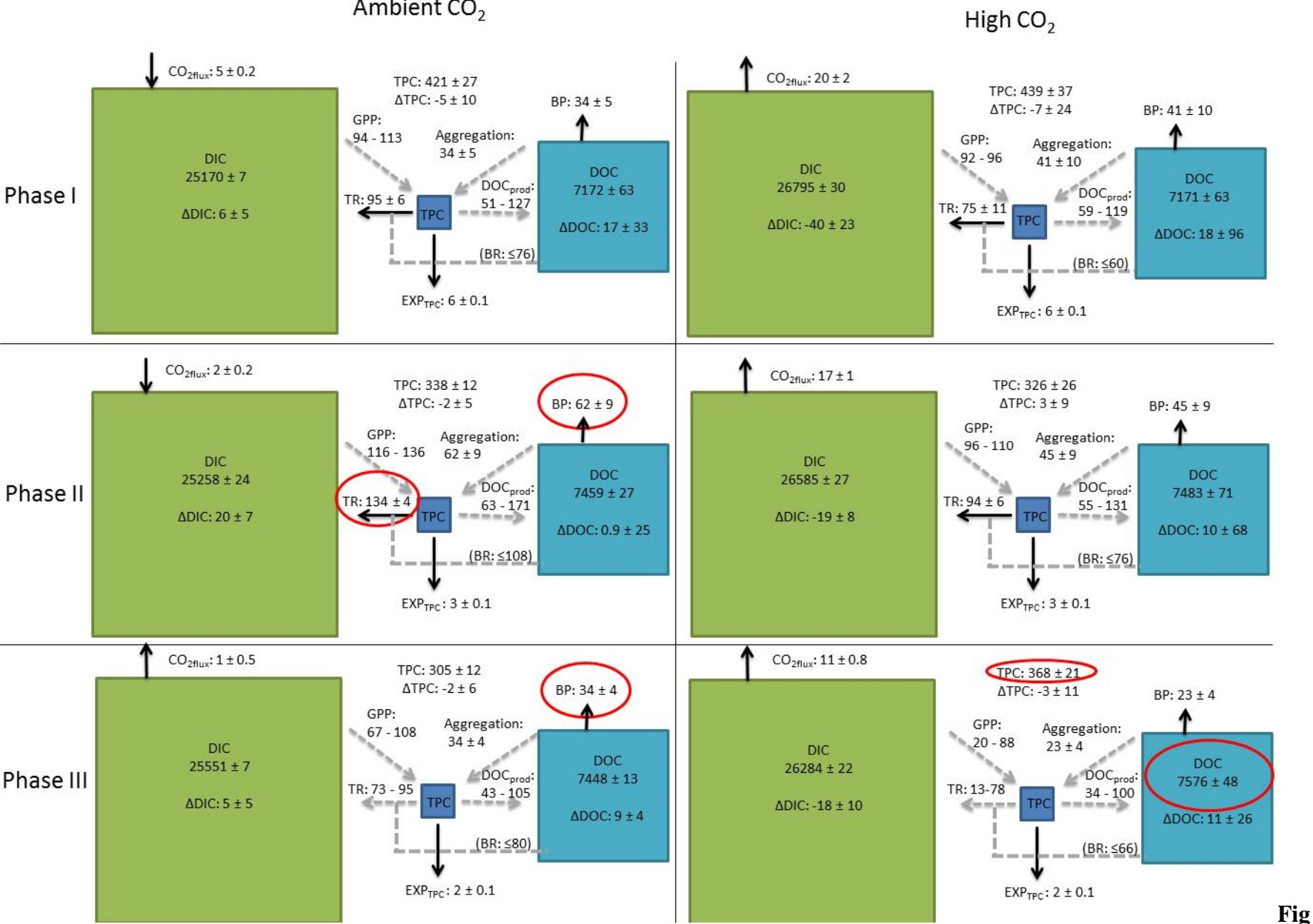

**Fig 5**

