# Peer review of "Effects of ocean acidification on pelagic carbon fluxes in a"

_Biogeosciences, 2016_

## Referee Comment (RC1) · Anonymous Referee #1 · 29 Mar 2016

Spilling et al. have done a very nice effort compiling a wide amount of results from a 44 d mesocosm study in the Baltic Sea and calculating the carbon fluxes under different ocean acidification scenarios. Despite the big amount of data the manuscript is concisely written, easy to read and shows clear conclusions. Among the different results they have determined standing stocks and temporal changes of total particulate carbon, dissolved organic carbon, dissolved inorganic carbon and particulate organic carbon (POC) of specific plankton groups, as well as carbon-$CO_2$ fluxes, sedimentation and biological rates (primary production, bacterial production and total respiration). The main results show that elevated $CO_2$ conditions increased total particulate carbon and the DOC pool due to a decrease in respiration and bacterial production at elevated $CO_2$ concentrations. I think that this is a very interesting result that needs to be discussed more deeply in the manuscript. I refer the authors to Hopkinson et al. 2010

and Teira et al. 2012 for information about decreases in phyto and bacterial respiration under high CO2 concentrations. Sobrino et al. 2014 can be also used as a reference related to downregulation of phytoplankton metabolism under high CO2, which might be an appropriate topic for the discussion of the manuscript.

Regarding the data analysis, I like the idea of using estimated instead raw data to make comparisons between variables or when observed values are not available. However the authors should also provide more information to complement or justify the usage of estimated vs. measured data. For example when comparing NPP14C and NPPe the authors only say that results "agree reasonably well" which is a very general contention for this paper. In addition, during Phase III, total respiration was not measured and the authors estimated TR based on the NPPe TR-1 and BP TR-1 ratios during Phase II. Information about their correlations during Phase II would be desirable to justify the estimation carried out during Phase III.

Finally, an specific equation for the estimation of bacterial respiration would be nice to see in the Methods.

Minor issues: - Line 234 days - Line 269 correlated to?? - Line 410. Revise sentence " The initial increase in the…." - Line 425 during - Fig. 1 filtration - Fig. 2. What about using similar units in the Y axis and legend (i.e. uatm??)

---

## Referee Comment (RC2) · Anonymous Referee #2 · 24 Apr 2016

The manuscript of Spilling et al. reports on a mesocosm experiment conducted in the Baltic Sea to test for effects of increased CO2 on plankton carbon fluxes. This manuscript is part of a special issue and specifically reports on estimated net community rates and their variation due to increased pCO2. Although I am convinced of the scientific relevance of this study, I am not convinced considering this budgeting exercice as a separate manuscript is highly relevant. Spilling et al. under revision in this special issue already reports on decreasing respiration rates at high CO2, causing higher Chla, TPC and DOC concentrations in the high CO2 treatments. The added value of the present manuscript is to estimate plankton rates that have not been directly measured (NPPe, but see later comment on this term; GPPi, but again see later comment; BR; DOC production). I would definitely recommend merging the two Spilling et al. papers to provide a more comprehensive overview of what happened during this

experiment.

If this suggestion is not followed, this manuscript, in my opinion, needs major revisions in order to improve its clarity and to discuss and criticise more deeply what has been found.

Estimates of DOC, TPC and DIC pools in mol C m-2: I was wondering for quite a while how these initial pools have been calculated and how the authors could provide an error estimate on a single sampling. I saw in the other Spilling et al. that these pools were actually averages of 3 sampling dates at the start of each phase. This must be clarified in the present manuscript. Also, how were integrated pools estimated: it is mentioned (and only for DIC, L136) in the ms that volumetric concentrations in per kg were converted using seawater density. Obviously, they were further multiplied by the considered depth. Please clarify.

Estimates of DOC, DIC and TPC rates of change: no information is provided on how these rates have been calculated. I believe these were calculated through linear regressions of each stock evolution during the considered phase. This must be clarified. Looking at Table 1 of the other Spilling et al., there are some discrepancies with rates presented here (e.g. Exp TPC of 7.4 in the first mesocosm compared to 6.6 in this paper, but this is also the case for other rates). Looking at the important errors associated with these rate estimates, it does seem like many slopes are not significantly different from 0. Please comment. In that case, how is it possible to compare these rates between the different mesocosms. Were these differences actually tested?

Estimates of NPPe and GPPi: Based on observed variations of TPC, DOC and DIC, the authors further calculated biological carbon fluxes. Net primary production measured by the 14C method (over 24h incubations) were compared to, what the authors refer to as NPPe being the missing process closing the organic budget: NPPe = Export + net variation in TPC + net variation in DOC). As the authors correctly mention, NPPe does incorporate total respiration and not only autotrophic respiration, as does

the 14C method (this is actually clearly doubtful considering the long incubations that have been performed)). Anyway, this is incorrect to refer to this process as Net Primary production, this is misleading and you really should consider using the proper term: Net Community Production, and as it is based on an organic budget, you should use NCPo. The authors further use an inorganic budget (based on DIC net fluxes, and estimated $CO_2$ fluxes) to estimate Gross Primary Production. I would strongly recommend for clarity to reconsider this part and to calculate NCPi, being the Net Community Production based on the inorganic budget. This is, I believe, what is shown in Fig. 3 and termed as Biological release or uptake. The authors have thus two estimates of the plankton community metabolism that do provide different outputs. While it seems like the inorganic budget shows that the community was heterotrophic in ambient mesocosms (Biological release of DIC), the organic budget suggests the opposite for all phases. This must be discussed. The paper as it stands is highly confusing with respect to this metabolic aspect. i.e. In the abstract is mentioned that during phase 1, the community under ambient and high $CO_2$ treatments was autotrophic (i.e. more production than respiration, with capacity to export to the sediment traps and export to the DOC pool). However, it is clearly stated that the community was heterotrophic during the entire experiment under ambient $CO_2$ conditions. Again, this must be clarified.

Comparison between the inorganic and organic budget: I already mentioned this, but I would like to insist on the fact that this paper reports on budgets based on both inorganic and organic constituents. Since they do not really agree, this must be deeply discussed. A recommendation on which type of budget is the most relevant and associated with the lowest uncertainties should be further proposed.

$CO_2$ effects on estimated rates: I do not see how differences between estimated rates between low and high $CO_2$ treatments have been tested. It is mentioned on L345 that "an effect of the different $CO_2$ treatments was noticeable in the NPPe but not in the NPP14C", how was it tested?

Comparison between NPP14C and NPPe: as correctly stated by the authors, NPP14C

rates should provide equal or higher estimates than NPPe (NCPo, see above). This is not the case and attributed (on top of potential errors on one control mesocosm) to "changed parameterisation during in incubation in small volumes". Based on my experience, we usually observe higher rates in small incubations vs large ones, not really in accordance with the lower rates of NPP14C during phase 1. Alternatively, this offset could be attributed to errors associated with NPPe estimates, since the TPC pool was clearly underestimated. Could you comment on this?

Estimates of biological pools: I do not really get what is the added value of calculating and presenting pools of meso- and microzooplankton, micro- and nanophytoplankton, picophytoplankton, bacteria and viruses. I would guess that these informations are already available in other manuscripts from the special issue. Is this not the case? This makes a small paragraph of the Results and Discussion and, apart from showing that measured TPC is much much lower than the cumulated stocks of these biological compartments, I do not see what valuable information it brings.

Estimates of variability: I would recommend the authors to mention the sample size each time SE are provided. Furthermore, I do not really see (as this is not explained) how SEs have been calculated for estimated rates (error propagation). e.g. as NPPe rates are based on DOC net fluxes, therefore the associated errors should be at least equal to the errors associated with DOC net fluxes right? This is not the case, and this must be clarified further.

Minor issues: Abstract: L57: did not transfer, please correct L58: revealed a clear effect of increasing CO2 on carbon production. I don't think this is correct. Carbon production does not seem impacted, while carbon loss is.

Materials and Methods: L104: I understood that more mesocosms were initially deployed, I do not see why this is not mentioned here. Everyone must also know how hard such experiment is. L159: Grossart et al. (2006), please correct L185: according to: , please correct L194: Cherny et al. (2013b), please correct. L205: organic carbon

pool, please add dissolved + particulate for clarity L207: Direct measurements using . . ., please correct

Results and Discussion: L265: While some indication on temporal evolution is provided for the other measured variables, this is not the case for bacterial biomass, please add this information. L280: Spilling et al. 2016), please correct L286: in e.g., please remove e.g. L287: have pointed at, please correct L297: p<= 0.01 I believe, please correct. L325: (Paul et al. 2015 (a or b)), please correct and clarify L353: Spilling et al. 2016), please correct

Figures Fig. 3. As mentioned earlier, for clarity, Biological release or uptake should be referred to as NCPi (based on the inorganic budget). Values are not in mol C m-2 but in mol C m-2 d-1 I think, please correct.

---

## Author Response (AR1)

**Response to review**

Dear Editor

We are grateful for all the constructive comments and suggestions by the reviewers, which have improved the manuscript a lot. Below we have placed all our original responses plus additional comments and changes made to the manuscript. Additional text put in the manuscript is marked in red.

On behalf of all the authors

Sincerely,

Kristian Spilling
* * *
**Reviewer #1, Comment #1**

The main results show that elevated CO2 conditions increased total particulate carbon and the DOC pool due to a decrease in respiration and bacterial production at elevated CO2 concentrations. I think that this is a very interesting result that needs to be discussed more deeply in the manuscript. I refer the authors to Hopkinson et al. 2010 and Teira et al. 2012 for information about decreases in phyto and bacterial respiration under high CO2 concentrations. Sobrino et al. 2014 can be also used as a reference related to downregulation of phytoplankton metabolism under high CO2, which might be an appropriate topic for the discussion of the manuscript.

**Author response:**

A good point and we will expand the discussion on this topic relating the decrease in respiration to possible downregulation of metabolism.

**Additional comments and Changes made:**

We did cover the reduced respiration and bacterial production in the Spilling et al 2016 paper and we do not want to be too repetitive, but we did expand on this in order to cover all relevant topics. We included the references suggested by the reviewer and expanded on the discussion around changes in the metabolism and possible impact on carbon concentrating mechanisms.

**Reviewer #1, Comment #2**

Regarding the data analysis, I like the idea of using estimated instead raw data to make comparisons between variables or when observed values are not available. However the authors should also provide more information to complement or justify the usage of estimated vs. measured data. For example when comparing NPP14C and NPPe the authors only say that results "agree reasonably well" which is a very general contention for this paper. In addition, during Phase III, total respiration was not measured and the authors estimated TR based on the NPPe TR-1 and BP TR-1 ratios during Phase II. Information about their correlations during Phase II would be desirable to justify the estimation carried out during Phase III.

**Author response:**

We will make changes to the estimated variables according to the suggestions of reviewer #2 (see comments below). We will be more specific when comparing different variables and also provide a better justification for the estimates of TR in Phase III. This was done using two methods as to give a range rather than specific number for the TR estimate.

**Change made:**

This was also a topic covered by reviewer #2 and we have followed those recommendations to make the distinction between the different properties easier to follow. For the comparison (referring to the statement "agree reasonably well") has now been removed. The details regarding the changes made can be found below under the comments made by reviewer #2.

66 Regarding the estimations carried out in Phase III, we are aware that this is highly uncertain, but this
67 is also why we did it using two different ways and this can be seen as a much wider span in the
68 estimate presented in Fig 5.

69

70

71 **Reviewer #1; Comment #3**

72 Finally, a specific equation for the estimation of bacterial respiration would be nice to

73 see in the Methods.

74

75 **Author response:**

76 This equation will be added

77

78 **Changes made:**

79

80 We added the equation in parenthesis as it can also be deduced from the following equation:  Eq 10

81

82 The added text: …(BR $\leq$ BB/TB)…

83

84

85 **Reviewer #1; Minor comments**

86 Minor issues: - Line 234 days - Line 269 correlated to?? - Line 410. Revise sentence

87 " The initial increase in the: : :." - Line 425 during - Fig. 1 filtration - Fig. 2. What about

88 using similar units in the Y axis and legend (i.e. uatm??)

89

90 **Author response:**

91 Appropriate changes will be made

92

93 **Changes made:**

94 All the changes were made with the exception of the comment to Fig 2. The legend has units µatm as
95 these are the treatments. The flux has been measured in mmol m-2 d-1.

96

97

**Reviewer #2, Comment #1**

Although I am convinced of the scientific relevance of this study, I am not convinced considering this budgeting exercice as a separate manuscript is highly relevant. Spilling et al. under revision in this special issue already reports on decreasing respiration rates at high CO2, causing higher Chla, TPC and DOC concentrations in the high CO2 treatments. The added value of the present manuscript is to estimate plankton rates that have not been directly measured (NPPe, but see later comment on this term; GPPi, but again see later comment; BR; DOC production). I would definitely recommend merging the two Spilling et al. papers to provide a more comprehensive overview of what happened during this experiment.

If this suggestion is not followed, this manuscript, in my opinion, needs major revisions in order to improve its clarity and to discuss and criticise more deeply what has been found.

**Author response:**

We do understand this point as having one manuscript was the original idea. During the writing process, however, we decided to present the budgeting exercise on its own in order to keep a more focused paper on respiration and primary production. The present manuscript was submitted as a synthesis paper and additionally presents data from many of the other papers submitted to the special issue, including bacterial production, DOC and a budget for the DIC based on atmospheric exchange. We are confident that following the referees' comments and suggestions will considerably improve our manuscript and justify separate publication.

**Changes made:**

We have taken the second approach and made a major revision, where we e.g. have reformulated some of the terms according the reviewer's suggestion (see comments below)

**Reviewer #2, Comment #2**

Estimates of DOC, TPC and DIC pools in mol C m-2: I was wondering for quite a while how these initial pools have been calculated and how the authors could provide an error estimate on a single sampling. I saw in the other Spilling et al. that these pools were actually averages of 3 sampling dates at the start of each phase. This must be clarified in the present manuscript. Also, how were integrated pools estimated: it is mentioned (and only for DIC, L136) in the ms that volumetric concentrations in per kg were converted using seawater density. Obviously, they were further multiplied by the considered depth. Please clarify.

**Author response:**

The reviewer is correct, the error estimates where made from consecutive measurements, and this will
be mentioned in the materials and methods and table legends (Tables 1-3). We will also add the
information that the depth and area of the mesocosms were used to calculate all pools and fluxes in m-
2 units.

**Changes made:**

We made a new paragraph under section '2.4. Data treatment', where these issues have been
addressed (inserted below). Part of this paragraph was mover up from the last paragraph of this
section for improving the readability.

"Based on the primary variables the experiment where divided into three distinct phases:
Phase I: *t0-t16*; Phase II: *t17-t30* and Phase III: *t31-t43*, where e.g. Chlorophyll *a* (Chl *a*)
concentration was relatively high during Phase I, decreased during Phase II and remained low
during Phase III (Paul et al. 2015). Measurements of pools and rates were average for the two
first sampling points of each experimental phase (n = 2) and where normalized to $m^2$
knowing the total depth and volume of the mesocosms. The three different phases of the
experiments were of different length (16, 14 and 13 days respectively). For fluxes and
biological rates we used the average for the whole periods normalized to days ($day^{-1}$) All
error estimates were calculated as standard error (SE), with n = 16, n = 14 and n = 13 for
Phases I – III respectively. SE for estimated rates were calculated from the square root of the
sum of variance for all the variables (Eq 6-11 below) The primary papers present detailed
statistical analyses and we only refer to those here. "

**Reviewer #2, Comment #3**

Estimates of DOC, DIC and TPC rates of change: no information is provided on how these rates have
been calculated. I believe these were calculated through linear regressions of each stock evolution
during the considered phase. This must be clarified. Looking at Table 1 of the other Spilling et al.,
there are some discrepancies with rates presented here (e.g. Exp TPC of 7.4 in the first mesocosm
compared to 6.6 in this paper, but this is also the case for other rates). Looking at the important errors
associated with these rate estimates, it does seem like many slopes are not significantly different from
0. Please comment. In that case, how is it possible to compare these rates between the different
mesocosms. Were these differences actually tested?

**Author response:**

This is a good point. It was calculated based on the difference between the start of each period, and using the average of the first two sampling days as the initial value for each period. So they are not slopes per se. There is no statistical testing of the differences in this paper, but we have explained that this was done in the paper where the original data is presented (and here linear regressions were used e.g. Paul et al 2015).

The discrepancy between the table in this paper with the other Spilling et al. paper is that here we did not include the time before the start of the $CO_2$ treatment (this will be changed also in the other Spilling et al paper), i.e. discarding the Exp TPC data from day T-1.

**Changes made:**

How these rates were calculated is now described in the new paragraph under section 2.4. Data treatment. Please see our response to the previous comment.

Changes to the Spilling et al. (2016) paper have been made.

**Reviewer #2, Comment #4**

Estimates of NPPe and GPPi: Based on observed variations of TPC, DOC and DIC, the authors further calculated biological carbon fluxes. Net primary production measured by the 14C method (over 24h incubations) were compared to, what the authors refer to as NPPe being the missing process closing the organic budget: NPPe = Export + net variation in TPC + net variation in DOC). As the authors correctly mention, NPPe does incorporate total respiration and not only autotrophic respiration, as does the 14C method (this is actually clearly doubtful considering the long incubations that have been performed)). Anyway, this is incorrect to refer to this process as Net Primary production, this is misleading and you really should consider using the proper term: Net Community Production, and as it is based on an organic budget, you should use NCPo. The authors further use an inorganic budget (based on DIC net fluxes, and estimated $CO_2$ fluxes) to estimate Gross Primary Production. I would strongly recommend for clarity to reconsider this part and to calculate NCPi, being the Net Community Production based on the inorganic budget. This is, I believe, what is shown in Fig. 3 and termed as Biological release or uptake. The authors have thus two estimates of the plankton community metabolism that do provide different outputs. While it seems like the inorganic budget shows that the community was heterotrophic in ambient mesocosms (Biological release of DIC), the organic budget suggests the opposite for all phases. This must be discussed. The paper as it stands is highly confusing with respect to this metabolic aspect. i.e. In the abstract is mentioned that during phase 1, the community under ambient and high $CO_2$ treatments was autotrophic (i.e. more production than respiration, with capacity to export to the sediment traps and export to the DOC pool). However, it is clearly stated that the community was heterotrophic during the entire experiment under ambient $CO_2$ conditions. Again, this must be clarified.

209

**Author response:**

We were a bit back and forth on how to best present the different variables when the manuscript was being written, and we ended up using the estimated net and gross production. The reviewer has a good point suggesting a better distinction between measured primary production and the estimated community production. We will change the NPPe to Net Community Production, organic budget (NCPo) and furthermore add the Net Community Production, inorganic budget (NCPi) as the reviewer suggests.

We will carefully go through the suggested points for clarification and discuss more in detail the discrepancy between the organic and inorganic carbon budget.

**Changes made:**

We changed the NPPe to become NCPo as suggested, and furthermore introduced the net community production based on inorganic carbon budget NCPi (Eq 6). This change was done throughout the manuscript including tables and figure legends.

We added a paragraph discussing the difference between NCPo and NCPi. They are different as the reviewer pointed out, but overall both give the same picture that NCP was higher in high CO2 treatments. Because of this we decided to change around a bit on the order of the Results and Discussion chapter, and placed the Biological rates after the presentation of the different carbon pools.

The new paragraph reads:

"The results of the DIC pool and atmospheric exchange of $CO_2$ provides another way of estimating the net community production based on inorganic carbon ($NCP_i$). There was some discrepancy between the $NCP_o$ and $NCP_i$ as the latter suggested net heterotrophy in the ambient $CO_2$ whereas the high $CO_2$ treatments were net autotrophic during all three phases of the experiment (Fig. 3). For the NCPo there was no indication of net heterotrophy at ambient $CO_2$ concentration. In terms of the absolute numbers, the NCPi estimate is probably more uncertain than $NCP_o$. Calculating the $CO_2$ atmospheric exchange from the measurements of a tracer gas involves several steps, each adding uncertainty to the calculation. However, both estimations (NCPi and NCPo) indicate that increased $CO_2$ concentrations lead to higher overall community production, supporting our overall conclusion. "

**Reviewer #2, Comment #5**

Comparison between the inorganic and organic budget: I already mentioned this, but I would like to insist on the fact that this paper reports on budgets based on both inorganic and organic constituents. Since they do not really agree, this must be deeply discussed. A recommendation on which type of budget is the most relevant and associated with the lowest uncertainties should be further proposed.

**Author response:**

This comment relates to comment #4 above and our reply to that. We will make the distinction between the organic and inorganic carbon budget as suggested and expand on this in the discussion.

**Changes made**

Please see our reply to Comment #4 above.

**Reviewer #2, Comment #6**

CO2 effects on estimated rates: I do not see how differences between estimated rates between low and high CO2 treatments have been tested. It is mentioned on L345 that "an effect of the different CO2 treatments was noticeable in the NPPe but not in the NPP14C", how was it tested?

**Author response:**

It was not tested statistically, and the term 'noticeable' refers to visual inspection of the data. We will however make a statistical test to strengthen this conclusion.

**Changes made:**

We added linear regression test to further underline this point and the line now reads:

There was no consistent difference between $CO_2$ treatments for $NPP_{14C}$ (p > 0.1), but $NCP_o$ increased with increasing $CO_2$ enrichment during Phase II (Phase II; linear regression p = 0.003; $R^2$ = 0.91).

**Reviewer #2, Comment #7**

Comparison between NPP14C and NPPe: as correctly stated by the authors, NPP14C rates should provide equal or higher estimates than NPPe (NCPo, see above). This is not the case and attributed (on top of potential errors in one control mesocosm) to "changed parameterisation during in

incubation in small volumes". Based on my experience, we usually observe higher rates in small incubations vs large ones, not really in accordance with the lower rates of NPP14C during phase 1. Alternatively, this offset could be attributed to errors associated with NPPe estimates, since the TPC pool was clearly underestimated. Could you comment on this?

**Author response:**

We do not have a good explanation for the discrepancy between NPPe and NPP14C, but underestimating of NPP14C seems more plausible as this are incubations in small volumes involving more steps than bulk measurements of TPC. Another possible explanation, suggested by the reviewer, is that the discrepancy could be due to an overestimation of NPPe. That would indicate an overestimation of either ΔTPC, ΔDOC or exported TPC (or a combination of these variables). TPC is not likely to be overestimated considering the methodology used, as measuring TPC has a relatively small uncertainty and would miss the <0.7 µm fraction. With this assumption, exported TPC would have been substantially overestimated. ΔTPC or ΔDOC would only be overestimated in the case when there is an underestimate at the start point, an overestimate at the end point or both an underestimate and overestimate increasing the difference between experimental phases in TPC or DOC. The discrepancy between NPPe and NPP14C during Phase I is so consistent for all treatments that we have hard time believing that we would have this consistent overestimation of ΔTPC or ΔDOC in all mesocosm bags.

Concerning the statement that: 'the TPC pool was clearly underestimated', we assume that you refer to the difference between the TPC pool and what was found in the bacterial and virus fraction based on flow cytometry. The small bacterial/virus not caught on the GFF filter did not contribute to the NPPe estimate. Defining the TPC as the >0.7 µm fraction, it is not obvious that TPC is underestimated.

**Reviewer #2, Comment #8**

Estimates of biological pools: I do not really get what is the added value of calculating and presenting pools of meso- and microzooplankton, micro- and nanophytoplankton, picophytoplankton, bacteria and viruses. I would guess that these informations are already available in other manuscripts from the special issue. Is this not the case? This makes a small paragraph of the Results and Discussion and, apart from showing that measured TPC is much much lower than the cumulated stocks of these biological compartments, I do not see what valuable information it brings.

**Author response:**

There is no paper presenting all of the organism groups together, and this being a synthesis paper we wanted to present these different pools. We agree that this data is not well incorporated into the story and we will expand on this in the discussion, trying to better link the relative contribution of the different groups to the fluxes presented.

317

**Changes made:**

We expanded on the community section (3.1.), relating the different plankton groups to the prevailing nutrient regime primarily driven by recycling. We also added a new paragraph about the virus fraction that was missing in the original manuscript.:

"Although there are some uncertainty in the carbon estimate (Jover et al. 2014), virus make up (due to their numerical dominance) a significant fraction of the pelagic carbon pool. Of the different plankton fractions the virioplankton have been the least studied, but their role in the pelagic ecosystem is ecologically important (Suttle 2007 NMR, Brussaard et al. 2008 ISMEJ; Mojica et al., 2016 ISMEJ). Viral lysis rates were equivalent to the grazing rates for phytoplankton and for bacteria in the current study (Crawfurd et al., 2015). As mortality agents they are drivers of the regenerative microbial food web, viruses (Suttle 2007, Brussaard et al. 2008). Overall, the structure of the plankton community reflected the nutrient status of the system. The increasing N-limitation favoring development of smaller cells, and increasing dependence of the primary producers on regenerated nutrients. "

**Reviewer #2, Comment #9**

Estimates of variability: I would recommend the authors to mention the sample size each time SE are provided. Furthermore, I do not really see (as this is not explained) how SEs have been calculated for estimated rates (error propagation). e.g. as NPPe rates are based on DOC net fluxes, therefore the associated errors should be at least equal to the errors associated with DOC net fluxes right? This is not the case, and this must be clarified further.

**Author response:**

We agree that the error estimates needs to be better explained. In the case of NPPe the SE was calculated from the square root of the sum of variance of the three parameters used to calculate the NPPe: DOC TPC and Exported TPC. We will include the sample size as suggested.

**Changes made:**

We added the sample size (n = ) for each SE in the text and tables. We also added the way SE was determined for the biological rates as described above in the materials and methods chapter and also in the table legend.

**Reviewer #2, Comment #10**

Minor issues: Abstract: L57: did not transfer, please correct L58: revealed a clear effect of increasing $CO_2$ on carbon production. I don't think this is correct. Carbon production does not seem impacted, while carbon loss is.

**Author response:**

This will be corrected.

**Changes made:**

L57 Corrected as suggested by the reviewer

L58 changed to: revealed a clear effect of increasing $CO_2$ on the carbon budget

**Reviewer #2, Comment #11**

Materials and Methods: L104: I understood that more mesocosms were initially deployed, I do not see why this is not mentioned here. Everyone must also know how hard such experiment is. L159: Grossart et al. (2006), please correct L185: according to: , please correct L194: Cherny et al. (2013b), please correct. L205: organic carbon pool, please add dissolved + particulate for clarity L207: Direct measurements using : : :, please correct

**Author response:**

We will refer to the Paul et al. (2015) paper where the initial deployments are mentioned and the overall methods are described in more detail and incorporate the suggested corrections.

**Changes made:**

We added the text: "… (nine KOSMOS units were originally deployed but three were lost due to leaks). A more detailed description of the set-up can be found in Paul et al. (2015)."

The rest of the changes were made according to the reviewer's suggestions

**Reviewer #2, Comment #12**

Results and Discussion: L265: While some indication on temporal evolution is provided for the other measured variables, this is not the case for bacterial biomass, please add this information. L280: Spilling et al. 2016), please correct L286: in e.g., please remove e.g. L287: have pointed at, please correct L297: p<= 0.01 I believe, please correct. L325: (Paul et al. 2015 (a or b)), please correct and clarify L353: Spilling et al. 2016), please correct

**Author response:**

This will be added and corrections will be made.

**Changes made:**

There was temporal development of the bacterial community and this information was added:

"Overall, the bacterial numbers largely followed the phytoplankton biomass with an initial increase then decrease during Phase I; increase during Phase II and slight decrease during Phase III (Crawfurd et al., 2016)."

All the other corrections were made according to the reviewer's suggestions

**Reviewer #2, Comment #13**

Figures Fig. 3. As mentioned earlier, for clarity, Biological release or uptake should be referred to as NCPi (based on the inorganic budget). Values are not in mol C m-2 but in mol C m-2 d-1 I think, please correct.

**Author response:**

The reviewer is right and this will be corrected.

**Changes made:**

This was corrected in Fig 3 and we also adopted the NCPi parameter instead of the biological release/uptake.

[revised manuscript text omitted]

DIC
25170 ± 7

ΔDIC: 6 ± 5

TPC: 421 ± 27
ΔTPC: -5 ± 10

GPP:
94 - 113

Aggregation:
34 ± 5

BP: 34 ± 5

TR: 95 ± 6

TPC

DOC$_{prod}$:
51 - 127

DOC
7172 ± 63

ΔDOC: 17 ± 33

(BR: ≤76)

EXP$_{TPC}$: 6 ± 0.1

CO$_{2flux}$: 20 ± 2

DIC
26795 ± 30

ΔDIC: -40 ± 23

TPC: 439 ± 37
ΔTPC: -7 ± 24

GPP:
92 - 96

Aggregation:
41 ± 10

BP: 41 ± 10

TR: 75 ± 11

TPC

DOC$_{prod}$:
59 - 119

DOC
7171 ± 63

ΔDOC: 18 ± 96

(BR: ≤60)

EXP$_{TPC}$: 6 ± 0.1

**Phase II**

CO$_{2flux}$: 2 ± 0.2

DIC
25258 ± 24

ΔDIC: 20 ± 7

TPC: 338 ± 12
ΔTPC: -2 ± 5

GPP:
116 - 136

Aggregation:
62 ± 9

BP: 62 ± 9

TR: 134 ± 4

TPC

DOC$_{prod}$:
63 - 171

DOC
7459 ± 27

ΔDOC: 0.9 ± 25

(BR: ≤108)

EXP$_{TPC}$: 3 ± 0.1

CO$_{2flux}$: 17 ± 1

DIC
26585 ± 27

ΔDIC: -19 ± 8

TPC: 326 ± 26
ΔTPC: 3 ± 9

GPP:
96 - 110

Aggregation:
45 ± 9

BP: 45 ± 9

TR: 94 ± 6

TPC

DOC$_{prod}$:
55 - 131

DOC
7483 ± 71

ΔDOC: 10 ± 68

(BR: ≤76)

EXP$_{TPC}$: 3 ± 0.1

**Phase III**

CO$_{2flux}$: 1 ± 0.5

DIC
25551 ± 7

ΔDIC: 5 ± 5

TPC: 305 ± 12
ΔTPC: -2 ± 6

GPP:
67 - 108

Aggregation:
34 ± 4

BP: 34 ± 4

TR: 73 - 95

TPC

DOC$_{prod}$:
43 - 105

DOC
7448 ± 13

ΔDOC: 9 ± 4

(BR: ≤80)

EXP$_{TPC}$: 2 ± 0.1

CO$_{2flux}$: 11 ± 0.8

DIC
26284 ± 22

ΔDIC: -18 ± 10

TPC: 368 ± 21
ΔTPC: -3 ± 11

GPP:
20 - 88

Aggregation:
23 ± 4

BP: 23 ± 4

TR: 13-78

TPC

DOC$_{prod}$:
34 - 100

DOC
7576 ± 48

ΔDOC: 11 ± 26

(BR: ≤66)

EXP$_{TPC}$: 2 ± 0.1

**Fig 5**

---

## Author Response (AR2)

**Response to review**

Dear Editor

I have gone through the reviewer's comments and fully agree that the description of the SE calculations has not been good enough. The confusion stems from a combination of a few mistakes in the 'n' number and the description of how it was calculated.

I have carefully gone through the manuscript and made appropriate corrections/changes, mainly in the materials and methods chapter and the Table and Figure legends. When going through the manuscript again I found a few minor things (e.g. spelling) that were also corrected in this version. A detailed response to each point is given below, plus the full manuscript with track changes.

Sincerely,

Kristian Spilling

**All the issues and points raised by the reviewer followed with our response**

I am afraid I still don't understand correctly as it seems that answers provided and the new paragraph that was added do not match. If I understand correctly, rates of change (i.e. Delta DOC) were calculated as the difference between the start (2 first days) of each period. My first question would be: how did you do for the last period?

**Author response** – it was done from the average of the last two days and this information has been added to the text (M&M chapter) and to the Table 3 legend.

In the added paragraph, it does not seem to be explained correctly. The same is true for the legends of Table 1-3, as it is said that: "… net community production estimated based on organic carbon pools (NCPo) are all average for Phase I in mmol C m-2 d-1 ± SE (n = 16)." How can you have n = 16 if based on a difference between 2 time points?

**Author response** – It was not based on the difference between two time points, as all variables were measured throughout the different phases, and the SE was calculated from the full set of data (please see also our explanation to your example of DOC below). You are right that the n = 16 is inaccurate. NCPo was calculated based on Eq. 5, for all the measured variables we have now placed the exact n number (it was 16 only for TR).

Moreover, regarding measured rates (i.e. BP), I doubt that they were measured on a daily basis (n is not 16), but I might be wrong.

**Author response** – You are partly right. NPP and TR were measured on a daily basis (but with a loss of some number of NPP as the incubation platform at some point disappeared), but other parameters including BP was measured every 2-3 days. In the figure legend we now specify the exact n for all variables.

If I take the example of Delta DOC in M1 for Phase 1, that would be 16.4 mmol C/m2/d (considering a 16 day period, 7435 for the start of phase 1 - 7172 for the start of phase 2), a value of 15.5 is reported. I end up with a NCPo estimate of 18.2 mmol C/m2/d.

**Author response** – Phase I, M1: the total period counted was from t0 to t16, which is 17 days when including day 0 as the first day, ending up with 15.5 instead of 16.4. This affects also NCPo.

More importantly, I don't understand that, if indeed you propagated errors the way you explained (i.e. square root of the sum of variance), how you end up with a propagated error that is lower than one of the terms. For instance, still for M1 in Phase 1, delta DOC should have an error of SQRT(87^2+38^2) = ca. 95. Doing the same for delta TPC (SE ~40), you would end up with a propagated error on NCPo of SQRT(95^2+40^2+0.1^2) = 103, far from the value of 33 that is reported.

**Author response** –The confusion most likely stems from an error in the n value set for the Delta (Δ) variables, and that the explanation of SE calculations was not elaborate enough. In your given example the n value is 8 and not 2. The SE for the Δ variables was not calculated from SE of the pools, e.g. $DOC_{pool}$ as in your example. Rather, the change was calculated between each measuring point and the SE for Δ variables calculated for the full range. We have corrected the n value in the figure legend, and elaborated the description in the materials and methods chapter.

I might have misunderstood, however if this is the case, first I apologize, but also I would suggest the authors to better clarify their methodology.

**Author response -** We fully agree and have hopefully managed to make the description of the SE calculations clear.

Very minor corrections:

L18-21: Affiliations have been swapped
**Author response** – Corrected

L50: Please remove "fixed"
**Author response** – removed

L180: A parenthesis is missing in the equation
**Author response** – corrected

L650: since you calculate GPP from your budgets, you should change to: (i.e. NCP + TR)
**Author response -** L650 is in the reference section, I looked through all places where GPP appeared, but did not find any apparent place where '(i.e. NCP + TR)' should be inserted.

[revised manuscript text omitted]